# Live imaging of wound angiogenesis reveals macrophage orchestrated vessel sprouting and regression

David B Gurevich[1], Charlotte E Severn[1,2], Catherine Twomey[1], Alexander Greenhough[3], Jenna Cash[1,4], Ashley M Toye[1,2,5] (iD), Harry Mellor[1] & Paul Martin[1,6,7,*] (iD)

## Abstract

Wound angiogenesis is an integral part of tissue repair and is impaired in many pathologies of healing. Here, we investigate the cellular interactions between innate immune cells and endothelial cells at wounds that drive neoangiogenic sprouting in real time and *in vivo*. Our studies in mouse and zebrafish wounds indicate that macrophages are drawn to wound blood vessels soon after injury and are intimately associated throughout the repair process and that macrophage ablation results in impaired neoangiogenesis. Macrophages also positively influence wound angiogenesis by driving resolution of anti-angiogenic wound neutrophils. Experimental manipulation of the wound environment to specifically alter macrophage activation state dramatically influences subsequent blood vessel sprouting, with premature dampening of tumour necrosis factor-α expression leading to impaired neoangiogenesis. Complementary human tissue culture studies indicate that inflammatory macrophages associate with endothelial cells and are sufficient to drive vessel sprouting via vascular endothelial growth factor signalling. Subsequently, macrophages also play a role in blood vessel regression during the resolution phase of wound repair, and their absence, or shifted activation state, impairs appropriate vessel clearance.

**Keywords** angiogenesis; inflammation; macrophages; neutrophils; wound
**Subject Categories** Development & Differentiation; Vascular Biology & Angiogenesis
**The EMBO Journal (2018) 37: e97786**

## Introduction

Healthy tissues rely on a plentiful vascular supply in order to receive the appropriate amount of oxygen and nutrients and to efficiently remove waste products. This reliance increases in instances of tissue damage where the healing process leads to intensive metabolic demands due to increased local cell signalling, migration and proliferation. Most tissue damage involves the disruption or destruction of the local vasculature, which must then be restored and increased in density to facilitate tissue repair. This process of new blood vessel growth from existing vessels, termed neoangiogenesis, must be tightly controlled; delayed or reduced neoangiogenesis may result in chronic non-healing wounds (Nunan *et al*, 2014), while over exuberant neoangiogenesis is associated with the formation of keloid scars and other forms of pathological fibrosis (Johnson & Wilgus, 2014). These two extreme outcomes of aberrant and dysregulated wound healing represent a substantial social and financial health burden (Sen *et al*, 2009). Therefore, understanding mechanisms of wound repair that might lead to improvements in wound care generally and management of wound neoangiogenesis in particular are highly relevant and desirable.

The process of wound healing progresses through a series of stereotypical, overlapping phases (Eming *et al*, 2014; Johnson & Wilgus, 2014; Cash & Martin, 2016). Innate immune cells play significant roles throughout these wound healing phases, mediating host defence, wound debridement, inflammatory status and signalling to coordinate cellular activities leading to appropriate migration, proliferation and tissue restoration (Eming *et al*, 2014; Martin & Nunan, 2015). Extensive studies of macrophages during developmental morphogenetic episodes have revealed that they associate with and support blood vessel sprouting and anastomosis in the hindbrain and retina of embryonic and neonatal mice (Fantin *et al*, 2010; Rymo *et al*, 2011), as well as the trunk vasculature in zebrafish embryos (Fantin *et al*, 2010). Some vessel regression during post-natal retinal

1  School of Biochemistry, University of Bristol, Bristol, UK
2  National Institute for Health Research (NIHR) Blood and Transplant Unit in Red Blood Cell Products, University of Bristol, Bristol, UK
3  School of Cellular and Molecular Medicine, University of Bristol, Bristol, UK
4  MRC Centre for Inflammation Research, Edinburgh Medical School, The Queen's Medical Research Institute, Edinburgh, UK
5  Bristol Institute for Transfusion Sciences, NHS Blood and Transplant, Filton, Bristol, UK
6  School of Physiology, Pharmacology and Neuroscience, University of Bristol, Bristol, UK
7  School of Medicine, University of Cardiff, Cardiff, UK
   *Corresponding author: Tel: +44 117 331 2298; E-mail: paul.martin@bristol.ac.uk

development is also driven by macrophages, as their absence completely abolishes this process (Lang & Bishop, 1993; Lobov *et al*, 2005; Rao *et al*, 2007). Furthermore, macrophages appear to be involved in both vascular sprouting and remodelling during normal physiological processes such as menstruation (Thiruchelvam *et al*, 2013) and at the maternal–foetal interface during pregnancy (Ning *et al*, 2016). These interactions can also be co-opted during pathological conditions; for example, the presence of pro-inflammatory macrophages is considered to be a key component of the dysregulated angiogenesis seen in psoriasis and atherosclerosis (Malecic & Young, 2017). Cancer growth and progression towards malignancy in mouse and man is generally associated with an increased local vasculature that is also heavily influenced by macrophage presence and phenotypic state (Lin *et al*, 2006; Williams *et al*, 2016), with some studies indicating a direct physical interaction between pro-angiogenic macrophages and cancer blood vessels (De Palma *et al*, 2005; Ojalvo *et al*, 2009). These direct, physical macrophage–vessel interactions have also been reported in other contexts, such as vessel damage observed in zebrafish models of stroke (Liu *et al*, 2016) and hypoxic stress (Gerri *et al*, 2017). However, macrophages are remarkably heterogeneous, and there is controversy as to whether pro- or anti-inflammatory phenotypes of these cells are responsible for the observed increases in vasculature (Squadrito & De Palma, 2011; Gerri *et al*, 2017). A likely key role for macrophages in wound neoangiogenesis is strongly indicated by their ablation at early time-points following murine wounding, which results in reduced vessel sprouting and aberrant wound angiogenesis (Lucas *et al*, 2010; Willenborg *et al*, 2012). However, the dynamic nature, precise function of the macrophage phenotypes involved, the exact timing of these interactions and what possible role other leucocytes such as neutrophils might be playing in this process is still unclear.

Here, we observe that following tissue wounding in both mice and zebrafish, macrophages largely recapitulate their role in development by associating with newly forming angiogenic sprouts, driving neoangiogenesis and subsequent remodelling of vessels at the repairing wound site. Utilising *in vivo* imaging of translucent transgenic zebrafish larvae in combination with genetic and pharmacological knockdown of macrophages, we capture the key events underlying the interactions between macrophages and blood vessels that effect wound neoangiogenesis, and demonstrate that wound inflammation is required for this process. Interestingly, as well as their direct role, macrophages also indirectly stimulate wound angiogenesis by driving resolution of wound neutrophils expressing factors that suppress angiogenesis. Switching to an *in vitro* human co-culture model, we show that it is specifically pro-inflammatory macrophages that are critical to initiating sprouting angiogenesis, and we return to the zebrafish to show that this action occurs via targeted delivery of pro-angiogenic cytokines including VEGF. Finally, we show a requirement for temporal phenotypic switching of the wound inflammation response, as the earlier activated pro-inflammatory macrophages must transition to an anti-inflammatory phenotypic state to permit appropriate later vessel remodelling and regression.

# Results

## Macrophages are associated with wound angiogenesis in mammals

Macrophages are thought to play key roles in developmental angiogenesis in both mouse and fish and appear to interact with blood vessel sprouts and anastomoses during the establishment of the embryonic vascular network (Fantin *et al*, 2010; Stefater *et al*, 2011). We reasoned that a similar association might be important in the context of wound angiogenesis since wound healing frequently recapitulates developmental processes (Martin & Parkhurst, 2004). Firstly, we characterised the wound neoangiogenic response in mice following a simple dorsal skin punch biopsy (Fig 1A) by immunostaining for the endothelial cell marker CD31 in wounds at timepoints throughout the repair process (Fig 1B). Our studies show that the first new blood vessels appear at 3 days post-injury (DPI) and are highly disorganised and poorly interconnected (Fig 1B and C). Wound blood vessels subsequently increase in number and branch pattern until they peak at 7 DPI, after which they resolve away from the wound area, becoming more organised and beginning to return to a distribution and alignment pattern resembling unwounded skin by 14 DPI (Fig 1B–D). This time course of events complements and expands on previous studies examining sprouting and regression of wound blood vessels using completely different technologies such as MicroCT angiography imaging (Urao *et al*, 2016).

▶

**Figure 1. Macrophages are associated with blood vessels in mouse wounds.**

A *En face* schematic illustration of a mouse wound, showing macrophage–blood vessel sprouts entering a full-thickness skin wound in relation to the overlying scab and associated inflammatory cells during wound healing.

B Representative multiphoton projection images of mouse wounds stained for endothelial cells (CD31) over the full duration of blood vessel infiltration and resolution.

C Quantification of total wound area occupied by blood vessels, based on pixel count (see Materials and Methods) and measured from time course represented in (B). Extent of wound vasculature increases to a maximum attained at 7 DPI, after which wound vasculature resolves towards uninjured levels by 14 DPI. *N* = 6 independent mice per timepoint.

D Quantification of blood vessel alignment, based on pixel orientation (see Materials and Methods) and measured from time course represented in (B). Vessels begin to restore their coherence and orientation by 14 DPI. *N* = 6 independent mice per timepoint.

E Representative multiphoton projection images of wound tissue co-stained for endothelial cells (CD31, red) and macrophages (CD68, green), showing stereotypical association of macrophages with blood vessel sprouts during vessel infiltration into wound site and their subsequent regression from the wound. Asterisks indicate macrophage–sprout interaction; arrows indicate phagocytosed vessel material within macrophages. These are matched with representative images of electron micrographs showing macrophage–endothelial cell interactions, taken at corresponding timepoints post-wounding. Macrophages are false coloured in green, with phagocytosed material in red.

F Quantification of blood vessel tip–macrophage association during the repair time course of the repair process, measured from the time course represented in (E). *N* = 5 independent mice per timepoint.

Data information: Error bars indicate mean ± SD. Scale bars: (B) = 200 μm, (E) = 10 μm.

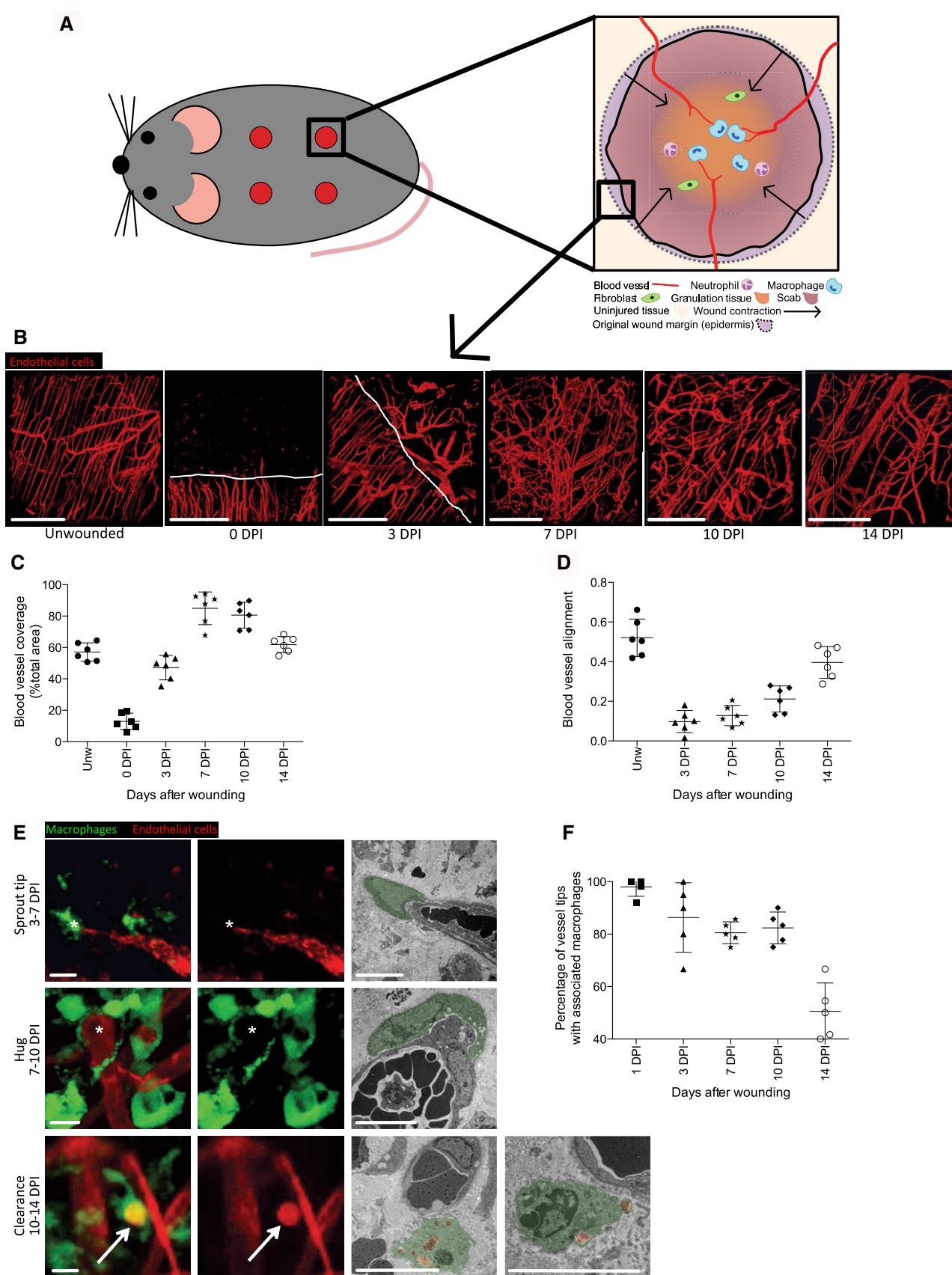

**Figure 1.**

We next examined for potential interactions between macrophages and blood vessels at various stages throughout the repair period. Co-immunostaining of excised wounds for macrophage and endothelial cell markers (CD68 and CD31, respectively) throughout wound repair and resolution phases revealed several typical interactions. Initially, during early and mid stages of wound repair from 3 to 7 DPI, macrophages appeared to be associated with the tips of newly formed neoangiogenic sprouts (Fig 1E and F). During later stages of wound repair, from 7 to 10 DPI, macrophages exhibited a shift in their associations with vessels, with less sprouting tip contact and a "hugging" behaviour with vessels that no longer appeared to be actively sprouting (Fig 1E). At the latest stages of repair, 10–14 DPI, some macrophages appeared to contain engulfed endothelial cell material (19.3 ± 3.8% of total macrophages) and this was confirmed by use of VE-Cadherin antibody as an alternative vessel marker (Figs 1E and EV1A and B, Movie EV1). We also observed each of these various types of macrophage–endothelial cell contacts at the ultrastructural level by transmission electron microscopy, which reveals details of such interactions. For example, in the period when vessel resolution is occurring, we see macrophages in the process of phagocytosing cell debris in close proximity to vessels (Fig 1E). Together, these observations suggest that the functions of macrophages in regulating angiogenesis may change throughout the wound repair time course, from nurturing and supporting endothelial cell sprouts during early stages of repair through to clearing of endothelial cells during later stages of wound resolution.

## Macrophages are associated with neoangiogenesis throughout the repair of zebrafish wounds

Our next objective was to directly observe these dynamic interactions between macrophages and blood vessels in living tissue. To this end, we developed several wounding protocols for zebrafish larvae, which are translucent and therefore more amenable to live imaging than the opaque tissues of mice. We characterised the time course of wound angiogenesis in zebrafish by utilising the Tg(*fli*:GFP) transgenic line, which marks all blood vessels. We first performed a relatively large mechanical needle-stick flank wound with a 30-gauge needle to the dorsal somite area opposite the cloaca and monitored the subsequent wound angiogenic response (Fig 2A). Zebrafish wound neoangiogenesis is temporally very similar to that of mammalian wounds, with the first blood vessel sprouts appearing within the wound at 1–3 DPI (Fig 2B). Unlike mature, undamaged vessels, these sprouts, and their resulting new vessels, are initially leaky, as evidenced by loading of blood vessels with high molecular weight rhodamine–dextran, which under unwounded conditions is too large to travel across the vessel wall (Fig EV2A and B). These new sprouts proceed to anastomose and form tortuous and densely branched vessels at 4–6 DPI (Fig 2B), by which stage they cease to leak loaded dextran (Fig EV2A and B). After reaching a post-wounding peak of complexity at about 6 DPI as measured by quantification of individual sprout and vessel node numbers, as well as total vessel length, these vessels subsequently resolve back to a pattern resembling that of uninjured tissue by 10 DPI (Fig 2B and C). Utilising a double-transgenic fish labelling arteries in green, Tg(*dll4*:GFP), and all vessels in red, Tg(*kdrl*:mCherry-CAAX), we were able to determine the nature of vessels involved in repair. In contrast to some studies of embryonic development where, for example, vessel sprouting and vascularisation of the spinal cord appear to proceed primarily from venous endothelium (Wild *et al*, 2017), we observed that both arteries and veins readily contribute to wound angiogenesis, resulting in chimeric wound vessels (contribution towards artery length ranged from 55.6% to 84.7%, while contribution towards vein length ranged from 15.3% to 44.4%, $n = 10$ independent fish at 4 DPI, Fig 2B). To determine whether physical vessel damage is required to stimulate

**Figure 2. Macrophages are associated with blood vessel sprouting tips in zebrafish wounds.**

A   Lateral schematic of the larger, 30-gauge needle-stick fish wounds made to the flank of larval zebrafish at 4 days post-fertilisation (DPF).

B   Representative confocal projection images of a wounded Tg(*fli*:GFP) zebrafish, showing the typical time course of blood vessel infiltration over 10 days post-injury following needle-stick injury as per (A). Boxed area denotes site of wounding. The Angioanalyser node/sprout quantification (see Materials and Methods) is generated for the entire area of interest and is visually indicated via the output from processing the 6 DPI image, showing nodes as red circles and sprouts as green lines. Representative confocal projection of a wounded Tg(*dll4*:GFP); Tg(*kdrl*: mCherry-CAAX) zebrafish, showing veins (red) and arteries (yellow), demonstrating a contribution to wound vasculature from both lineages with formation of chimeric vessels at the wound site.

C   Graphical representation of total vessel length, number of nodes and sprouts throughout the period of needle-stick wound repair as represented in (B), plotted against vessel measurements from uninjured fish. $N = 12$ independent fish per timepoint per condition (unwounded versus wounded).

D   Representative confocal projection images of Tg(*mpeg*:mCherry); Tg(*mpx*:GFP) transgenic zebrafish, showing time course of macrophage (red) and neutrophil (green) migration to needle-stick wounds as per (A), 0-7 DPI. Boxed area denotes site of wounding.

E   Quantification of macrophage and neutrophil numbers at the wound site versus uninjured tissue, measured from time course represented in (D). $N = 12$ independent fish per timepoint, per condition.

F   Representative confocal projection image of Tg(*fli*:GFP); Tg(*mpeg*:mCherry) or Tg(*fli*:GFP); Tg(*mpx*:GFP) transgenic zebrafish at 3 DPI, showing typical immune cell–blood vessel sprout interactions following needle-stick injury. Boxed area denotes site of immune cell–endothelial cell interaction.

G   Quantification of macrophage–blood vessel sprout association throughout early stages of repair, measured from time course represented in (F). $N = 16$ independent fish per timepoint.

H   Lateral schematic of smaller microlaser wound, performed on 4 DPF zebrafish.

I   Representative confocal projection images taken from timelapse movies of a laser injured Tg(*fli*:GFP); Tg(*mpx*:GFP); Tg(*mpeg*:mCherry) transgenic zebrafish, 4 DPF, imaged at 30–930 MPI. Neutrophils appear at the wound site early and transiently; macrophages appear at approximately the same time as blood vessel sprouting commences and remain associated with the repairing blood vessels throughout sprouting and anastomosis. Boxed area denotes wound site.

J   Representative confocal projection images taken of laser wounded Tg(*fli*:GFP) transgenic zebrafish injected with high molecular weight dextran (red) immediately before imaging. This "angiography" technique reveals how wound blood vessels are initially leaky, but that patency is restored once the vessel is repaired and lumenised. Boxed area denotes wound site. $N = 12$ independent fish per timepoint.

K   Graphical temporal representation of common cellular interaction events, measured from movies represented in (I). $N = 7$ independent fish.

Data information: For all graphs, error bars indicate mean ± SD. Scale bars: (B) 40 μm; (D) 100 μm; (F, I, J) 20 μm.

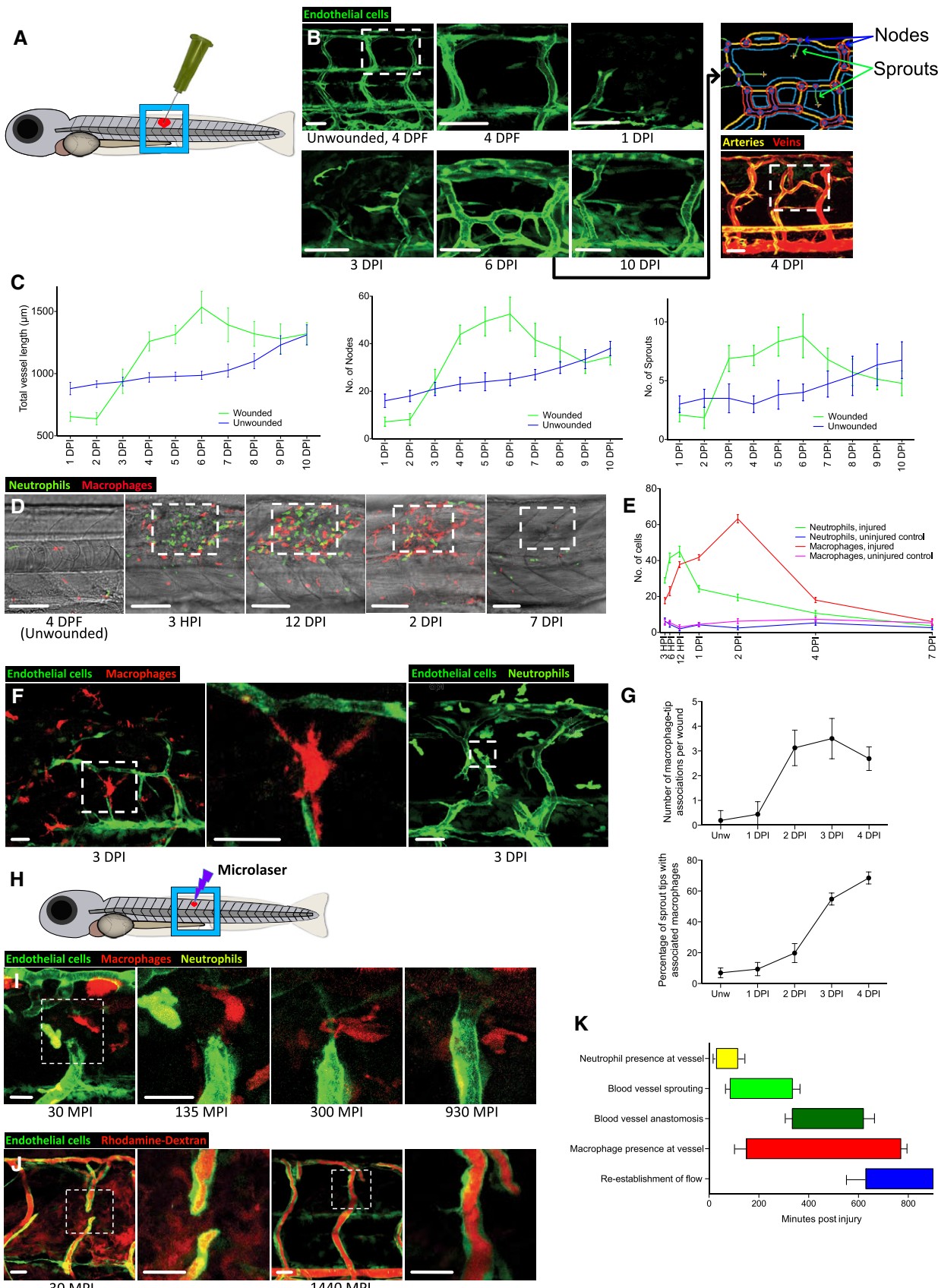

**Figure 2.**

wound neoangiogenesis, we performed smaller scale mechanical injuries using a fine tungsten needle, targeting somitic tissue but without direct damage to any vessel. Even without vessel damage, these wounds attracted new vessel sprouts, indicating that signalling from a non-endothelial exogenous source such as immune cells is responsible for driving wound neoangiogenesis (Fig EV2C).

Utilising transgenic fish lines with labelled neutrophils Tg(mpx: GFP) and macrophages Tg(mpeg:mCherry), we next examined the response of innate immune cells to a 30-gauge needle-stick injury (Fig 2D and E). Neutrophil recruitment to the wound peaks at 12 HPI and then resolves back to levels observed in uninjured tissue by 4 DPI (Fig 2D and E). Macrophages take longer to reach their peak in the wound area, at about 2DPI, and also longer to resolve, with some cells still present up to 10 DPI and beyond (Fig 2D and E). These invading waves of neutrophils and later macrophages at the site of injury are similar to those reported in previous wound studies (Gray *et al*, 2011). In combination with the Tg(fli:GFP) transgenic line, we examined interactions between blood vessels and both of these recruited immune lineages at the wound site. Neutrophils are rapidly drawn to the vicinity of wound blood vessels, but are only ever transiently observed in direct contact with sprouting tips and soon leave the immediate vicinity of wound angiogenesis. By contrast, macrophages form direct intimate associations with newly formed sprouting tips at early repair timepoints (Fig 2F and G), similar to what we observe in mouse wounds. These results indicate that the stereotypical interactions between macrophages and blood vessels observed during mammalian wound angiogenesis are largely conserved in zebrafish wounds.

To better analyse these cellular interactions over the full time course of vessel repair and resolution, we performed a second, smaller type of lesion by microlaser ablation (Fig 2H); these resolve in a shorter time window, thus enabling imaging for the full duration of repair. We performed microlaser ablations of trunk blood vessels of Tg(fli:GFP); Tg(mpx:GFP); Tg(mpeg: mCherry) triple transgenic fish and timelapse imaged the subsequent repair process. Small, partial intersegmental vessel (ISV) ablations allowed for imaging of the entire repair process through to vessel patency, while larger total ISV ablations allowed for a more extensive examination of immune cell–vessel interaction during sprouting. We observed that vessel sprouting at the wound site appears to follow a similar process to that reported for embryonic development of the ISVs (Isogai *et al*, 2001, 2003), with rapid filopodial extension and sprout growth. If the extent of vessel loss is less than 30 μm (small ablation), then vessel outgrowth generally occurs in a ventral to dorsal direction and leads to anastomosis and lumenisation by approximately 600 min post-injury (Fig 2I, Movie EV2). For full ISV ablations where vessel damage is greater than 60 μm, filopodial extension and sprout formation occur from both the dorsal aorta (DA) and dorsal longitudinal anastomotic vessel (DLAV); vessel tips extend filopodia in a highly dynamic fashion apparently sensing directionality before meeting and undergoing fusion at a midpoint between the two major vessels (Fig EV2D, Movie EV3). Movies of partial ISV ablations confirmed that neutrophil interactions with damaged blood vessels occur early and are generally transient (between 30 and 120 min post-injury), after which neutrophils generally had no further association with the vessel repair process

(Fig 2I–K, Movie EV2). By contrast, interactions between macrophages and damaged blood vessels persist for the full duration of vessel repair (Fig 2I–K, Movies EV2 and EV4). Macrophages initially localise to the gap between the two damaged vessel tips, after which they often wrap entirely around and "hug" the repairing blood vessel during the sprouting and early anastomosis phases (approx. 300 min post-injury); these intimate contacts are generally maintained until after vascular patency is restored (approx. 750 min post-injury; Fig 2I and J, Movies EV2 and EV4). To confirm these observations, we performed complementary experiments on Tg(kdrl:mCherry-CAAX); Tg(mpx:GFP); Tg(mpeg: mCherry) fish (Fig EV2E, Movie EV5). These live-imaging experiments strongly indicate important roles for macrophages that change during wound angiogenesis and its subsequent regression.

## Macrophages are required for neoangiogenesis in zebrafish wounds

To examine whether interactions between macrophages and blood vessels at the site of damage were directly facilitating wound neoangiogenesis, we utilised the metronidazole–nitroreductase system to specifically ablate macrophages. The double-transgenic line Tg (mpeg:Gal4FF); Tg(UAS:NfsB-mCherry), which drives nitroreductase enzyme via the Gal4-UAS system under the control of the macrophage mpeg promoter and enables efficient ablation of macrophages after treatment with metronidazole (Palha *et al*, 2013). We observe > 90% macrophage ablation within 48 h of exposure to metronidazole, with no detectable associated developmental defects (Fig EV3A). Recovery of metronidazole-treated fish by transfer into fresh water post-injury led to a partial restoration of macrophage numbers at the wound site at 4 DPI, by comparison to continually treated fish (Fig EV3C–E).

In the context of wound neoangiogenesis, ablation of macrophages prior to injury and throughout the entire first four days of repair lead to a complete failure of wound neoangiogenesis (Fig 3A and B). To complement this chemical-genetic approach, we used a pharmacological method for depleting macrophages involving the injection of toxic clodronate liposomes, an ablation regime previously utilised in mice (Van Rooijen & Sanders, 1994) and zebrafish regenerative studies (Carrillo *et al*, 2016; Figs 3A and B, and EV3B). This alternative macrophage ablation method resulted in similar retardation of wound angiogenesis (Fig 3A and B). Short-term metronidazole treatment up to the moment of injury followed by recovering the fish in fresh water during the injury repair period resulted in a partial blockade of wound angiogenesis, with some new vessels present at 4 DPI (Fig EV3C–E). Together, these results confirm that macrophages are required for neoangiogenesis at the wound and indicate that their impact on the angiogenic process is proportional to the extent of the macrophage response.

To more precisely live image the differences in vessel sprouting in the presence or absence of macrophages, we generated microlaser ablations of entire individual blood vessels in either Tg(fli:GFP); Tg (mpx:GFP); Tg(mpeg:mCherry) transgenic fish or Tg(fli:GFP); Tg (mpx:GFP); Tg(mpeg:Gal4FF); Tg(UAS:NfsB-mCherry) transgenic fish, treated with metronidazole as a control or to ablate macrophages throughout the repair process, respectively (Fig 3C, Movies EV6 and EV7). Macrophage ablation resulted in poorly directed and unsupported filopodial sprouting from leading tip cells, and a failure

of sprout extension and anastomosis. We also observed that in the absence of macrophages, neutrophils are retained at the wound site and maintain long-term associations with damaged blood vessel tips rather than the transient interactions that occur in wounds with an intact macrophage response (Fig 3C–E, Movie EV7). Conversely, we observed contact-dependent neutrophil displacement from vessel tips driven by macrophages in the unablated situation, similar to that seen previously in other wound scenarios (Tauzin *et al*, 2014). To test whether these neutrophils might be inhibitory to wound angiogenesis, we performed quantitative reverse transcription–polymerase

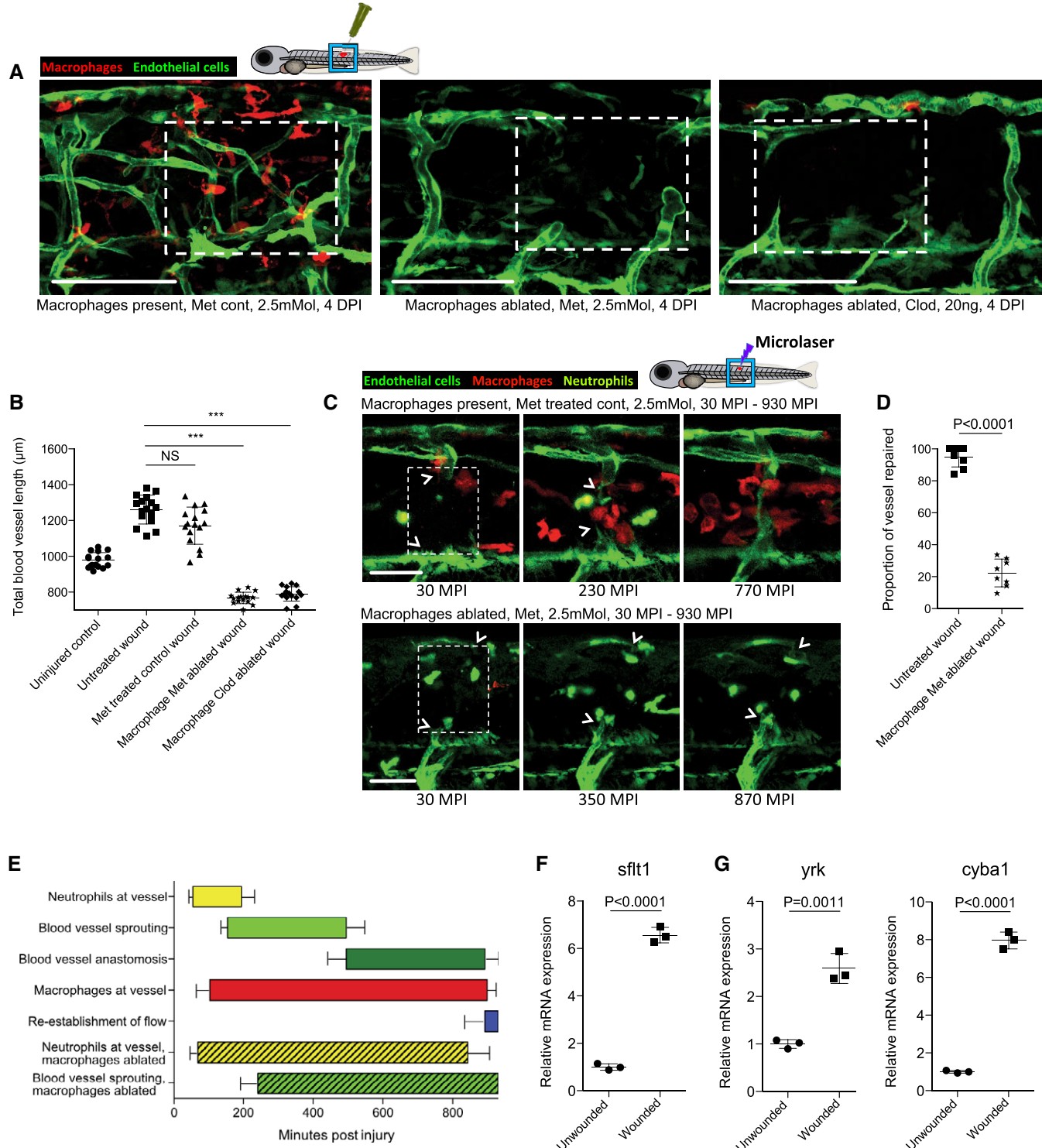

**Figure 3.**

chain reaction (quantitative PCR) on FAC-sorted neutrophils from early stab wounds; these neutrophils exhibit increased expression of anti-angiogenic genes relative to those from unwounded fish (Fig 3F). Specifically, early wound neutrophils express a soluble, truncated version of vegf receptor 1, *sflt1*, which has a high affinity for vegf but lacks a membrane domain (Kendall & Thomas, 1993). We find that it is only this truncated version of vegf receptor 1 that is upregulated and that primers designed specifically against the membrane version of *flt1* indicated a lack of expression in neutrophils (Fig EV3F). Expression of *sflt1* has previously been implicated in fine-tuning vegf signalling during developmental angiogenesis, being expressed by vessel stalk cells and thereby acting as a localised vegf sink, resulting in maximised vegf signalling and guidance to vessel tip cells (Chappell *et al*, 2009). We further show that early wound macrophages express increased levels of *cyba* and *yrk*, a pair of redox-regulated Src family kinase genes normally associated with appropriate macrophage recruitment and subsequent neutrophil reverse migration (Tauzin *et al*, 2014; Fig 3G). These results, together with our *in vitro* data (see later), indicate that early wound neutrophils may provide an initial angiogenic brake and that macrophages are involved in dispersing neutrophils to remove this brake, as well as positively directing vascular sprouts at the wound site by supporting and stabilising their initial outgrowth as they progress to vessel anastomosis.

## The activation state of macrophages at the wound site influences the extent of neoangiogenesis in zebrafish wounds

Tissue inflammation at the site of injury is a typical early step in wound repair. Since macrophages are known to switch phenotypic state during the repair period (Novak & Koh, 2013), we reasoned that this may influence their capacity to regulate various aspects of wound neoangiogenesis and subsequent vessel resolution throughout the wound repair process. To observe the number of pro-inflammatory, activated macrophages at the wound site over time, we utilised the Tg(*tnfα*:GFP); Tg(*mpeg1*:mCherry) double-transgenic

fish, since tumour necrosis factor-α (tnfα) is a known marker of the pro-inflammatory phenotypic state (Parameswaran & Patial, 2010; Marjoram *et al*, 2015). After tissue wounding, we observed a wave of pro-inflammatory macrophages constituting more than 50% of the total macrophages at the wound at 1 DPI but decreasing to almost none by 4 DPI (Fig 4B–E). In order to manipulate the inflammatory state of wound macrophages, fish were treated immediately after wounding with either lipopolysaccharide (LPS) as a non-specific immunostimulant (Fig EV4A) or with recombinant zebrafish interferon gamma (Ifn-γ) as a more specific immunostimulant (Fig 4A and B). In another series of wounded fish, we suppressed wound inflammation using hydrocortisone (Fig EV4A) or recombinant zebrafish interleukin-10 (Il-10) (Fig 4A and B). These contrasting treatments allowed us to manipulate macrophage phenotype either towards or away from a *tnfα*-positive, pro-inflammatory phenotype (Fig 4A). As expected, we observed an increase in the proportion of *tnfα*-positive, inflammatory wound macrophages when treated with either immunostimulant, versus a decrease in proportion of inflammatory wound macrophages when treated with either immunosuppressive agent (Figs 4B, D and E, and EV4A–C). We next examined how the modulated wound inflammatory response impacted wound neoangiogenesis. Our experiments revealed a substantial decrease in the level of blood vessel sprouting and a general impairment of wound neoangiogenesis following treatment with either of the immunosuppressive agents for needle-stick wounds (Figs 4B and F, and EV4A and D). Conversely, the degree of wound neoangiogenesis was generally increased when treated with either immunostimulatory agent (Figs 4B and F, and EV4A and D). However, injection of high molecular weight dextran into LPS-treated fish, and to a lesser extent Ifn-γ-treated fish, indicated that the newly established blood vessels at these wounds were leakier than those found at equivalent stages in untreated wounds at the same stage post-wounding (4 DPI; Fig EV4E).

To manipulate wound macrophage phenotype using complementary genetic approaches, we also wounded fish mutant for colony-stimulating factor 1 receptor a (*csf1ra*). *csf1ra* mutant zebrafish are

---

**Figure 3. Macrophages are necessary for wound angiogenesis and act to stabilise wound blood vessel sprouting tips.**

A Representative confocal projection images of needle-stick injured Tg(*fli*:GFP); Tg(*mpeg*:mCherry) or Tg(*fli*:GFP); Tg(*mpeg*:KalTA4); Tg(*UAS*:nfsB-mCherry) zebrafish in combination with various treatments (as indicated), for ablating macrophages (red) during the peak stage of vessel (green) sprouting (4 DPI). Boxed area denotes wound site. In control (DMSO vehicle) and metronidazole-treated fish lacking nitroreductase expressing macrophages, a normal inflammatory response and wound angiogenesis is observed. Upon ablation of macrophages using the nitroreductase–metronidazole system, wound angiogenesis is almost completely blocked. The same is true when macrophages are ablated using clodronate liposome injections.

B Quantification of total blood vessel length in the presence/absence of macrophages at the wound site, measured from images represented in (A), using Angioanalyser. *N* = 16 independent fish per condition. Statistical significance, as determined by one-way ANOVA, is *P* ≤ 0.0001. Subsequent Bonferroni multiple comparison test, determines level of significance, as indicated. Significance values: ****P* ≤ 0.0001.

C Images taken from timelapse movies of 4 DPF laser wounded Tg(*fli*:GFP); Tg(*mpx*:GFP); Tg(*mpeg*:mCherry) control or Tg(*fli*:GFP); Tg(*mpx*:GFP); Tg(*mpeg*:KalTA4); Tg(*UAS*:nfsB-mCherry) macrophage ablation transgenic zebrafish, treated with metronidazole and imaged 30–930 min post-injury (MPI). Boxed area denotes wound site. Arrowheads indicate position of damaged, sprouting vessel tips.

D Quantification of proportion of vessel repaired within 930 MPI, measured from movies represented in (C). *N* = 8 independent fish per condition. Statistical significance is indicated, as determined by two-tailed *t*-test.

E Graphical representation of typical cellular interaction events and their time course, measured from movies represented in (C). *N* = 8 independent fish per condition.

F Gene expression of FAC-sorted neutrophils extracted from Tg(*mpx*:GFP); Tg(*mpeg*:mCherry) needle-stick wounded fish at 12 HPI, showing relative gene expression for *sflt1* compared to unwounded fish at 4.5 DPF. *N* = 50 independent fish per condition per replicate, 3 replicates. Statistical significance is indicated, as determined by two-tailed *t*-test.

G Gene expression of FAC-sorted macrophages from Tg(*mpx*:GFP); Tg(*mpeg*:mCherry) transgenic zebrafish at 12 HPI post-needle-stick injury, showing relative gene expression for *cyba1* and *yrk* compared to unwounded fish at 4.5 DPF. *N* = 50 independent fish per condition per replicate, 3 replicates. Statistical significance is indicated, as determined by two-tailed *t*-test.

Data information: For all graphs, error bars indicate mean ± SD. Scale bars: (A) 100 μm; (C) 40 μm

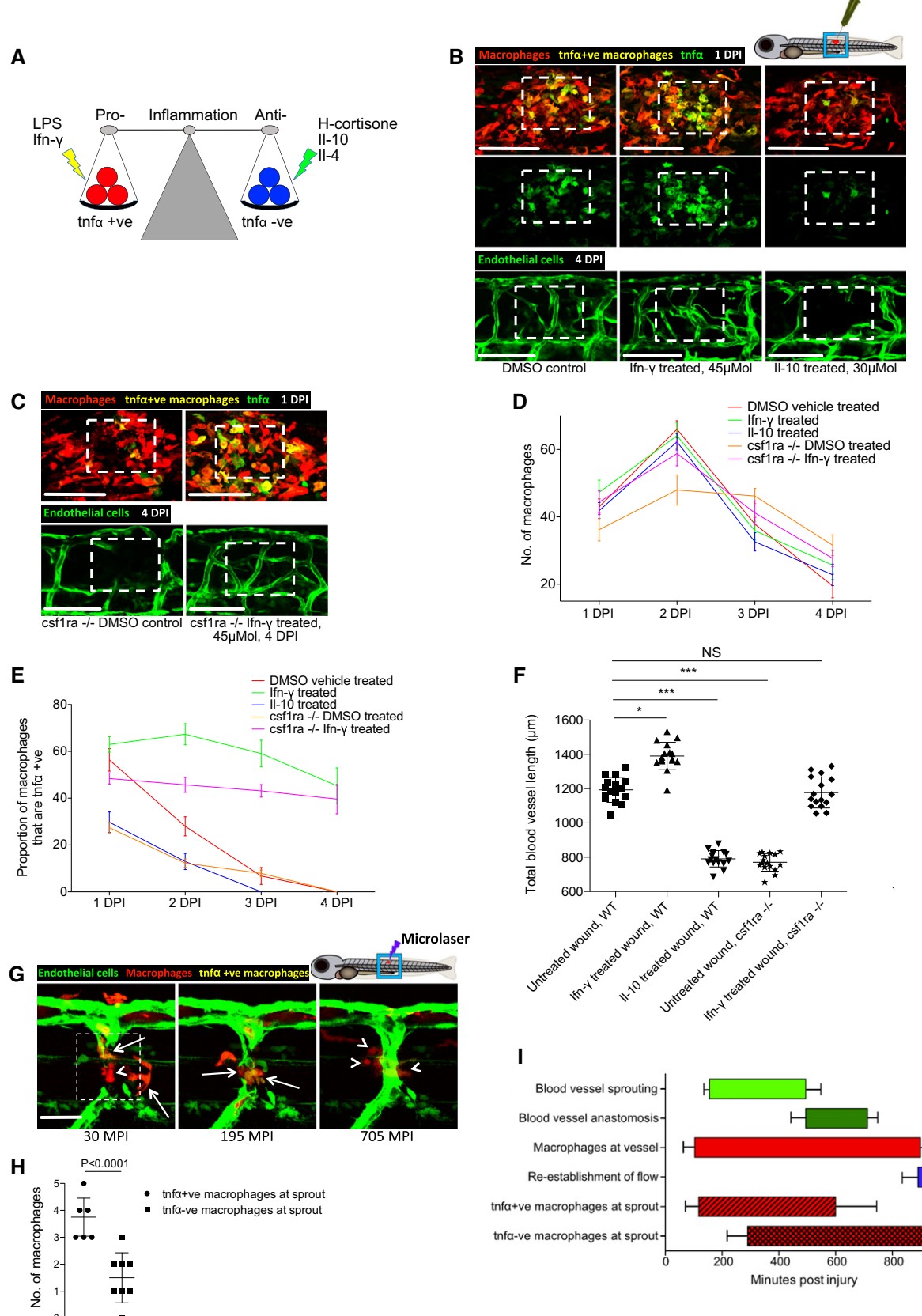

**Figure 4.**

**Figure 4.    Manipulation of wound inflammation and macrophage phenotypic state affects the extent of wound angiogenesis.**

A    Schematic showing the factors used to skew macrophage phenotype towards either pro-inflammatory or anti-inflammatory states.
B    Representative confocal projection images of needle-stick injured zebrafish, taken from either 1 DPI Tg(*mpeg*:mCherry); Tg(*tnfα*:GFP) transgenic zebrafish (top panel) or 4 DPI Tg(*fli*:GFP) transgenic zebrafish (bottom panel), treated, as indicated, with DMSO (control), Ifn-γ or Il-10. Boxed area denotes wound site.
C    Representative confocal projection images of needle-stick injured zebrafish taken from either 1 DPI Tg(*mpeg*:mCherry); Tg(*tnfα*:GFP) transgenic fish (top panel; middle panel for GFP only) or 4 DPI Tg(*fli*:GFP) transgenic fish (bottom panel), *csf1ra*⁻/⁻ mutant and treated with Ifn-γ. Boxed area denotes wound site.
D    Quantification of number of macrophages at the wound site between 1 and 4 DPI, measured from images represented in (B and C). N = 14 (from B) and N = 16 (from C) independent fish per timepoint per condition.
E    Quantification of proportion of macrophages that are *tnfα* positive at the wound site from 1 to 4 DPI, measured from images represented in (B and C). N = 14 (from B) and N = 16 (from C) independent fish per timepoint per condition.
F    Quantification of total blood vessel length at 4 DPI, measured from images represented in (B and C), using Angioanalyser. N = 14 (from B) and N = 16 (from C) independent fish per timepoint per condition. Statistical significance, as determined by one-way ANOVA, is $P \leq 0.0001$. Subsequent Bonferonni multiple comparison test, determines level of significance, as indicated. Significance values: *$P \leq 0.05$, ***$P \leq 0.0001$.
G    Representative confocal projection images taken from laser wounded Tg(*fli*:GFP); Tg(*mpeg*:mCherry); Tg(*tnfα*:GFP) triple transgenic zebrafish, imaged 30-930 MPI. Boxed area denotes wound site. *tnfα*-positive, inflammatory macrophages (arrows) associate with damaged, sprouting blood vessels earlier and in larger numbers than *tnfα*-negative, non-inflammatory macrophages (arrowheads).
H    Quantification of numbers of inflammatory versus non-inflammatory macrophages associated with damaged, sprouting blood vessels, measured from images represented in (G). N = 8 independent fish. Statistical significance is indicated, as determined by two-tailed *t*-test.
I    Graphical representation of common cellular interaction events and their time course, measured from movies represented in (G) and compared to events observed in Fig 3C. N = 8 independent fish.

Data information: For all graphs, error bars indicate mean ± SD. Scale bars: (B, C) 100 μm; (G) 40 μm.

known to have deficiencies in raising an appropriate immune response to bacterial infection (Pagan *et al*, 2015). While this mutant line exhibited macrophage recruitment to the wound site, it displayed a deficiency in its pro-inflammatory response, with considerably reduced macrophage *tnfα* expression (Figs 4C–E, and EV4A–C). We further observed that wound angiogenesis was inhibited in these pro-inflammation deficient fish (Figs 4C and F, and EV4A and D). Treating *csf1ra* mutants with immunostimulants LPS or Ifn-γ throughout the wound healing process rescued both the wound inflammation tnfα response as well as the wound angiogenesis deficit (Figs 4C and F, and EV4A and D). Taken together, these data indicate that wound inflammation status in general, and the switching of macrophages to a pro-inflammatory phenotype in particular, is critical in promoting wound angiogenesis during the early stages of wound repair.

Finally, laser ablation of single vessels in Tg(*fli*:GFP); Tg(*tnfα*: GFP); Tg(*mpeg*:mCherry) triple transgenic fish revealed that *tnfα*-positive macrophages interacted earlier and more frequently with laser damaged blood vessel tips than did *tnfα*-negative, non-inflammatory macrophages (Fig 4G–I, Movie EV8). Complementary movies of vessel repair following injuries performed on Tg(*kdrl*:mCherry-CAAX); Tg(*tnfα*:GFP); Tg(*mpeg*:mCherry) triple transgenic fish further demonstrated the rapid responsiveness of *tnfα*-positive macrophages to damaged vessel tips (Fig EV4F, Movie EV9). Additionally, observing multiple such movies revealed that some of these early responding *tnfα*-positive macrophages exhibited reduced GFP over time in the vicinity of others that remained bright, thus excluding photobleaching as an explanation; this suggests that *tnfα* expression by macrophages is transient and that at least some cells are switching from *tnfα* positive to *tnfα* negative during the repair period (Fig EV4F, Movie EV9). In the context of the overall vessel repair response, the presence of these *tnfα*-positive inflammatory macrophages temporally coincides with the early sprouting events, whereas non-inflammatory macrophages dominate later, as vessels undergo anastomosis (Fig 4G–I). These observations suggested that different macrophage phenotypes might be regulating different aspects of the wound angiogenesis process.

## Inflammatory and non-inflammatory macrophages play different roles in driving wound angiogenesis

The concept of a pro-inflammatory environment potentially acting as a trigger for vessel sprouting has been previously suggested (Sainson *et al*, 2008). To examine the cellular interactions underlying this observation more closely and to better understand what role they may be playing in driving neoangiogenesis, we established an *in vitro* co-culture assay using human umbilical vein endothelial cells (HUVECs), which when cultured together with feeder human fibroblasts spontaneously generated vessels in culture (Bishop *et al*, 1999; Richards & Mellor, 2016). To this assay, we added primary human macrophages isolated from peripheral blood mononuclear cells that had been treated with cytokines to direct them towards pro-inflammatory (interferon γ, IFN-γ) or anti-inflammatory (interleukin-4, IL-4) activation states (Fig 5A). We confirmed that macrophages had indeed reached differential activation states by testing for levels of gene expression of well-established pro-inflammatory (tumour necrosis factor-α, TNF-α) or anti-inflammatory (mannose receptor C-type 1, MRC1) genes (Roszer, 2015; Fig EV5A). To complement these co-culture cell-based approaches, we also exposed HUVECs to conditioned media extracted from the *in vitro* macrophage cultures, to dissect any differential contributions of cell-to-cell contact versus secreted factors produced by macrophages in these altered states (Fig 5A). The resultant macrophage–endothelial cell interactions were imaged and quantified after 4 days of co-culture. We observed three different types of macrophage–vessel interactions *in vitro*—macrophages can interact with blood vessel tips, or localise to the sides of blood vessels, or to vessel intersections and anastomoses. Intriguingly, we observed that pro-inflammatory macrophages were far more likely to associate with vessel tips than any other part of the vessel, whereas non-inflammatory cells (treated with IL-4) associated randomly with different areas of vessels (Fig 5B and C).

We next quantified the extent of vessel sprouting in these *in vitro* conditions. Our results showed that, just as *in vivo* in the zebrafish, the presence of pro-inflammatory macrophages (or their conditioned media) led to a significant increase in the amount and complexity of

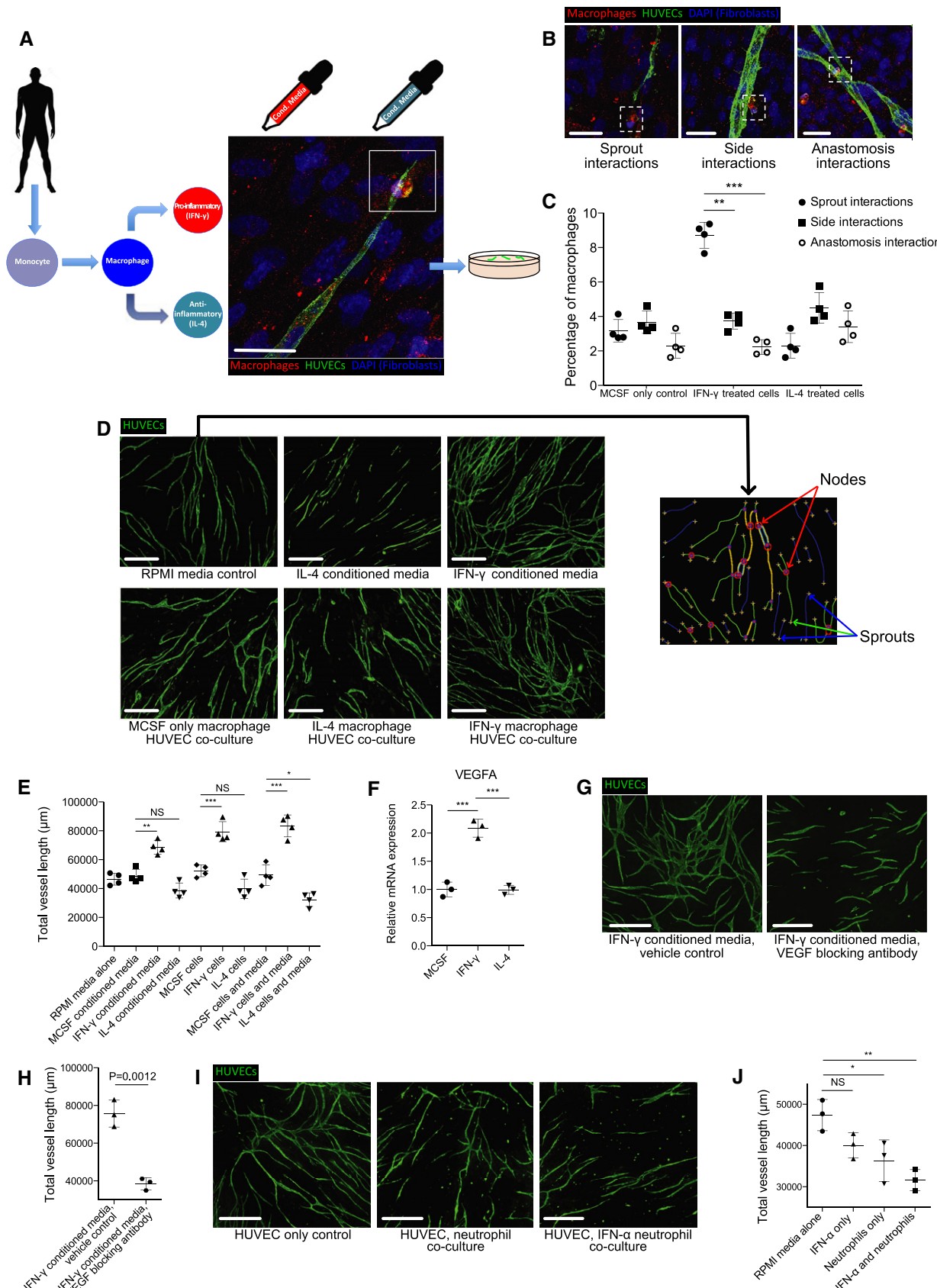

**Figure 5.**

**Figure 5. Inflammatory macrophages drive blood vessel sprouting in human HUVEC/macrophage co-culture via vegf signalling, while neutrophils suppress angiogenesis.**

A Schematic representing experimental design for human macrophage/human endothelial cell co-culture study. Primary monocytes are matured to macrophages, which are in turn skewed towards a pro-inflammatory status or an anti-inflammatory status by exposure to INF-γ or IL-4, respectively. These cells, or their media, are then added to HUVECs cultured on a feeder layer of fibroblasts.

B Representative confocal projection images taken from a human macrophage/HUVEC co-culture stained for endothelial cells (CD31, green), macrophages (MCSFR, red) and DAPI (nuclei, including those feeder cells), showing three types of observed cellular interactions: (i) macrophage–sprout interactions, (ii) macrophage interactions with the sides of vessels, (iii) macrophage interactions at the site of vessel anastomosis.

C Graphical representation of percentage of macrophage–HUVEC for each of the categories of interaction described in (B). $N = 4$ independent experiments per condition. Statistical significance, as determined by one-way ANOVA, is $P < 0.0001$.

D Representative fluorescent images of HUVECs co-cultured with macrophages or cultured in macrophage-conditioned media, as indicated, stained for endothelial cells (CD31). Culturing HUVECs in inflammatory macrophage-conditioned media or with inflammatory macrophages results in enhanced vessel sprouting, formation and growth.

E Quantification of total vessel length of HUVECs cultured or co-cultured as indicated, measured from images represented in (D) using Angioanalyser. $N = 4$ independent experiments per condition. Statistical significance, as determined by one-way ANOVA, is $P \leq 0.0001$.

F VEGFA expression levels for human macrophages treated with IFN-γ versus IL-4. $N = 3$ independent macrophage culture experiments per condition. Statistical significance, as determined by one-way ANOVA, is $P \leq 0.0001$.

G Representative fluorescent images of HUVECs cultured in conditioned media from IFN-γ-treated macrophages and either treated with vehicle control or VEGF blocking antibody, stained for endothelial cells (CD31, green).

H Quantification of total vessel length of HUVECs, measured from images represented in (G) using Angioanalyser. $N = 3$ independent experiments. Statistical significance is indicated, as determined by two-tailed $t$-test.

I Representative fluorescent images of HUVECs co-cultured with freshly isolated neutrophils, or neutrophils treated with IFN-α and stained for endothelial cells (CD31, green). Culturing HUVECs with neutrophils results in suppressed vessel sprouting, formation and growth.

J Quantification of total vessel length of HUVECs cultured or co-cultured with neutrophils as indicated, measured from images represented in (I) using Angioanalyser. $N = 3$ independent experiments per condition. Statistical significance, as determined by one-way ANOVA, is $P = 0.005$.

Data information: For all ANOVA tests, subsequent Bonferroni multiple comparison test was performed to determine level of significance, as indicated. Significance values: *$P \leq 0.05$, **$P \leq 0.001$, ***$P \leq 0.0001$. For all graphs, error bars indicate mean ± SD. Scale bars: (A, B) 20 μm; (D, G, I) 500 μm.

blood vessels in the co-culture (Figs 5D and E, and EV5B). By contrast, the presence of non-inflammatory macrophages (or their conditioned media) trended to an overall decrease in resultant blood vessels (Figs 5D and E, and EV5B). Similar increases and decreases in terms of vessel sprouting were observed when using LPS as a pro-inflammatory mediator, or dexamethasone as an anti-inflammatory mediator, respectively (unpublished observations). Quantitative PCR on macrophages cultured in immunostimulating conditions revealed high levels of VEGFA, suggesting that inflammatory macrophages might exert their function as angiogenic stimulants by providing a source of vascular growth factor (Fig 5F), complementing previous studies describing similar macrophage-derived VEGF as a driver for angiogenesis in mice (Xiong *et al*, 1998; Ramanathan *et al*, 2003; Cattin *et al*, 2015). Indeed, addition of VEGFA blocking antibody to cultures with pro-inflammatory macrophages reduced the amount of vessels formed to levels observed in control macrophage co-cultures (Fig 5G and H). These results suggest that pro-inflammatory macrophages play a specific role in initiating and driving vessel sprouting, at least in part, via release of VEGFA.

Since our earlier *in vivo* studies suggested that another pro-angiogenic role of macrophages might be to dislodge inhibitory neutrophils from vessel tips, we sought to examine this neutrophil anti-angiogenic activity using our co-culture *in vitro* assay. We isolated primary human neutrophils and cultured them, with or without pro-inflammatory interferon-α (IFN-α; Pylaeva *et al*, 2016; Shaul *et al*, 2016), together with HUVECs. Quantifying the extent of *in vitro* vessel sprouting indicated that neutrophils suppressed vessel growth, with IFN-α-treated neutrophils significantly more inhibitory (Fig 5I and J). These results suggest that neutrophils can indeed exhibit inhibitory angiogenic influences and thus may require dislodging at sites of *in vivo* angiogenesis.

To investigate further whether *tnfα*-positive, pro-inflammatory macrophages were indeed delivering pro-angiogenic cues at the wound site, we FAC-sorted tnfα-positive versus tnfα-negative macrophages from zebrafish wounds and performed quantitative PCR for a range of angiogenic genes. Our results confirmed that *tnfα*-positive macrophages express higher levels of *vegfaa* (Fig 6A). Quantitative PCR performed on whole zebrafish wounds in the context of macrophage ablation showed greatly reduced levels of *vegfaa* expression, suggesting that wound macrophages are indeed necessary for *vegfaa* expression at the wound site (Fig 6B). These observations were supported by whole-mount *in situ* hybridisation (WISH) for *vegfaa* performed on wounded Tg(*tnfα*:GFP); Tg(*mpeg*:mCherry) transgenic fish, which showed a marked increase in *vegfaa* levels at the wound site, particularly by *tnfα*-positive macrophages (Fig 6C). Furthermore, performing WISH for *vegfaa* on metronidazole-treated Tg(*mpeg*:Gal4FF); Tg(*UAS*:NfsB-mCherry) transgenic fish revealed a decrease in *vegfaa* staining at wounds where macrophages were absent (Fig 6C).

To genetically test the importance of *vegfaa* signalling, we utilised the *vegfaa^bns1* mutant zebrafish, which has severe developmental vascular defects but can be rescued via *vegfaa* RNA injection (Rossi *et al*, 2016). Injuring *vegfaa* null "rescued" larvae, indicated that, while they were able to raise a typical immune response to wounding, they failed to undergo wound angiogenesis (Fig 6D and E), suggesting that *vegfaa* is indeed critically important for vessel repair. To more precisely control the timing of vegf signal blockage, we treated needle-stick injured Tg(*fli*:GFP) fish with two vegfr2 inhibitors SKLB1002 and SU5416 which have previously been shown to inhibit developmental sprouting angiogenesis in zebrafish (Serbedzija *et al*, 1999; Zhang *et al*, 2011); treating larvae with either of these inhibitors completely blocked wound angiogenesis compared to vehicle-treated wounded controls (Figs 6F and G, and EV6A and B). Live imaging of laser ablations of entire vessels in Tg (*fli*:GFP); Tg(*mpeg*:mCherry) transgenic fish treated with SKLB1002 showed that while macrophages were still attracted to damaged

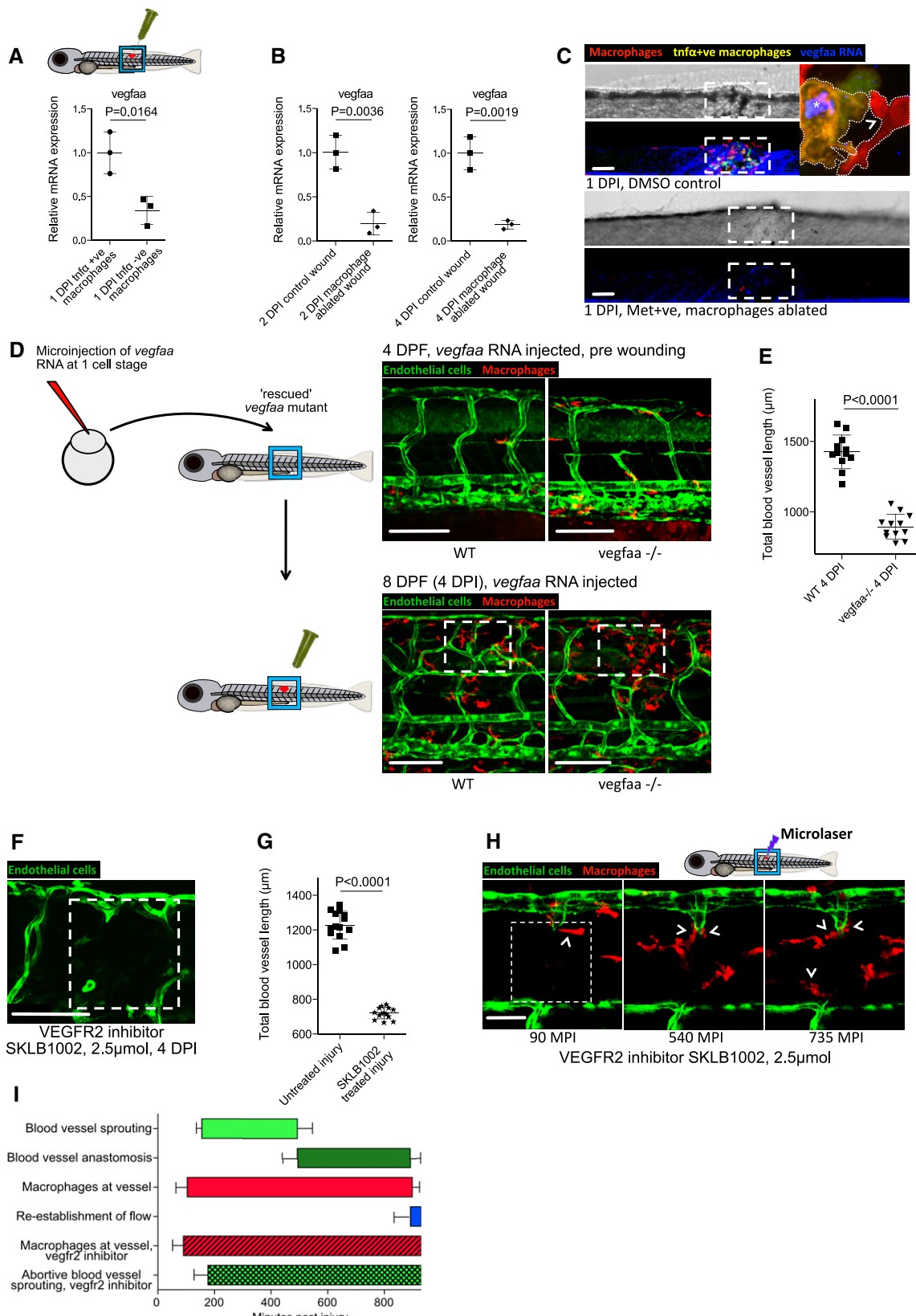

**Figure 6.**

◀ **Figure 6.  Zebrafish wound angiogenesis is driven by wound inflammatory macrophage-mediated vegf signalling.**

A   Gene expression for FAC-sorted zebrafish wound macrophages from Tg(*tnfα*:GFP); Tg(*mpeg*:mCherry) zebrafish, extracted at 1 DPI following needle-stick injury. *tnfα*-positive inflammatory macrophages express higher levels of *vegfaa* than *tnfα*-negative non-inflammatory macrophages. N = 50 independent fish per condition, 3 replicates. Statistical significance is indicated, as determined by two-tailed *t*-test.

B   Gene expression for whole-dissected zebrafish needle-stick wounds from Tg(*mpeg*:KalTA4); Tg(*UAS*:nfsB-mCherry) zebrafish following treatment with metronidazole, versus control DMSO-treated wounded fish, at 2 DPI and 4 DPI. Macrophage absence from wounds results in an overall decrease in *vegfaa* levels at the wound site. N = 30 independent fish per condition, three replicates. Statistical significance is indicated, as determined by two-tailed *t*-test.

C   Representative confocal projection images taken from needle-stick wounded Tg(*mpeg*:KalTA4); Tg(*UAS*:nfsB-mCherry); Tg(*tnfα*:GFP) triple transgenic zebrafish, DMSO control or metronidazole treated indicated, upon which *vegfaa* WISH was performed. Inset image shows high levels of *vegfaa* expression (blue) in *tnfα*-positive macrophages (yellow, asterisk) versus low levels in *tnfα*-negative macrophages (red, arrowhead). Boxed area denotes wound site. N = 12 independent fish per condition.

D   Schematic and representative confocal images showing rescue experiment and subsequent needle-stick injury performed on *vegfaa* homozygous mutant zebrafish or siblings, transgenic for Tg(*etv2*:GFP). Embryos derived from an incross of *vegfaa*$^{+/-}$ adults were injected with "rescue" mRNA encoding for vegfaa-165 at the one-cell stage. At 4 DPF, these mutant fish had an impaired but functional complement of ISVs. Needle-stick injuries performed on these mutants and their siblings at 4 DPF indicated a failure of wound angiogenesis by 4 DPI, despite a relatively normal inflammatory response. Boxed area denotes wound site.

E   Quantification of total blood vessel length at 4 DPI, measured from images represented in (D), using Angioanalyser. N = 12 independent fish per condition. Statistical significance is indicated, as determined by two-tailed *t*-test.

F   Representative confocal projection image taken from needle-stick wounded Tg(*fli*:GFP) transgenic zebrafish, imaged 4 DPI and treated with vegfr2 inhibitor SKLB1002 from the moment of injury. Boxed area denotes wound site.

G   Quantification of total blood vessel length, measured from images represented in (F) using Angioanalyser. N = 14 independent fish. Statistical significance is indicated, as determined by two-tailed *t*-test.

H   Representative confocal projection images taken from laser wounded Tg(*fli*:GFP), Tg(*mpeg*:mCherry) double-transgenic zebrafish, imaged 30–930 MPI and treated with SKLB1002 from moment of injury. Macrophages still associate with damaged vessels (arrowheads), but these vessels do not sprout appropriately and fail to repair. Boxed area denotes wound site.

I   Graphical representation of common cellular interaction events and their time course, measured from movies represented in (H) and compared to events observed in Fig 3C. N = 8 independent fish.

Data information: For all graphs, error bars indicate mean ± SD. Scale bars: (C, D, F) 100 μm; (H) 40 μm.

blood vessel tips, these tips failed to sprout and grow to any significant degree (Fig 6H and I, Movie EV10). Together, these results indicate that appropriate macrophage-driven inflammation, particularly in early stages of wound repair, is required to drive subsequent angiogenesis via a *vegf*-mediated mechanism and this may involve a close interaction of macrophages with vessel tips.

## Blood vessel resolution at the wound site is dependent on macrophage presence and phenotypic state

At later stages of wound repair, blood vessels at wounds are pruned back to uninjured levels because the reduced metabolic demand in the repaired tissues no longer necessitates such an extensive vascular network. Since our later stage murine wounds indicate that many wound macrophages contain material that stained for the CD31 endothelial cell marker (Fig 1D), we investigated whether this observation was also true in our zebrafish wounds. Indeed, imaging later stage wounds in Tg(*fli*:GFP); Tg(*mpeg*:mCherry) double-transgenic fish revealed macrophages containing fluorescent blood vessel material (1.2 ± 0.7% of total macrophages) (Fig 7A). To investigate whether macrophages are required for wound vessel resolution, we utilised the Tg(*fli*:GFP); Tg(*mpeg*:Gal4FF); Tg(*UAS*: NfsB-mCherry) triple transgenic fish and performed metronidazole-treated macrophage ablation at later stages than previously, from 6 DPI to 10 DPI, to delete macrophages only during this window of vessel regression as indicated in Fig 2B and C. Ablation of macrophages at this later stage resulted in considerably more vessels at 10 DPI than in untreated fish (Fig 7B and C). To test whether macrophage regulation of vessel resolution might be mediated by endothelial cell apoptosis, we treated the wounded fish with the pan-caspase inhibitor, QVD. Blocking all caspase activity in this manner produces phenotypes that mimic those seen upon late stage macrophage ablation (Fig 7B and C). We further analysed the

degree of apoptosis in vessels at the wound site using a combination of transgenic lines that both mark blood vessels, Tg(*kdrl*:mCherry-CAAX), and reveal apoptosis, Tg(*secA5*:YFP) (van Ham *et al*, 2010). Apoptosis was evident in endothelial cells within the vessel network at the wound site, as well as in some cells adjacent to these vessels (Fig 7D and E). By comparison, vessels in equivalent stage unwounded fish exhibited little if any apoptosis (Fig 7D and E). Combining this apoptosis detection approach with clodronate liposome injection and subsequent macrophage ablation at these later repair stages allowed us to examine the role of macrophages in this vessel remodelling process. These experiments showed that the absence of macrophages resulted in a decreased level of endothelial cell apoptosis (Fig 7D and E), similar to what has been demonstrated in vessel remodelling during mammalian growth (Lang & Bishop, 1993). However, some vessels were still capable of undergoing apoptosis, suggesting that a further macrophage independent mechanism may govern programmed vessel death in regressing vessels. Interestingly, FAC-sorted macrophages taken from these later stage wounds failed to express wnt7b and ang2 (unpublished observations), two factors that have previously been reported to be part of the pathway used by macrophages to induce endothelial cell apoptosis (Lobov *et al*, 2005; Rao *et al*, 2007). Together, these results confirm that macrophages play a critical role in regression of blood vessels at wound sites, by inducing some level of apoptosis in endothelial cells via an as yet unknown process, as well as by subsequent phagocytosis of apoptotic endothelial cells.

While observing these later stages of wound repair in Tg(*tnfα* GFP); Tg(*mpeg*:mCherry) double-transgenic fish, we observed a small number of macrophages that retain a *tnfα*-positive, inflammatory phenotype compared to unwounded controls where there are none (Fig 7F). By treating these wounded fish with the immunostimulant Ifn-γ from 6 DPI, we were able to increase the proportion of inflammatory macrophages at the repairing wound site, without

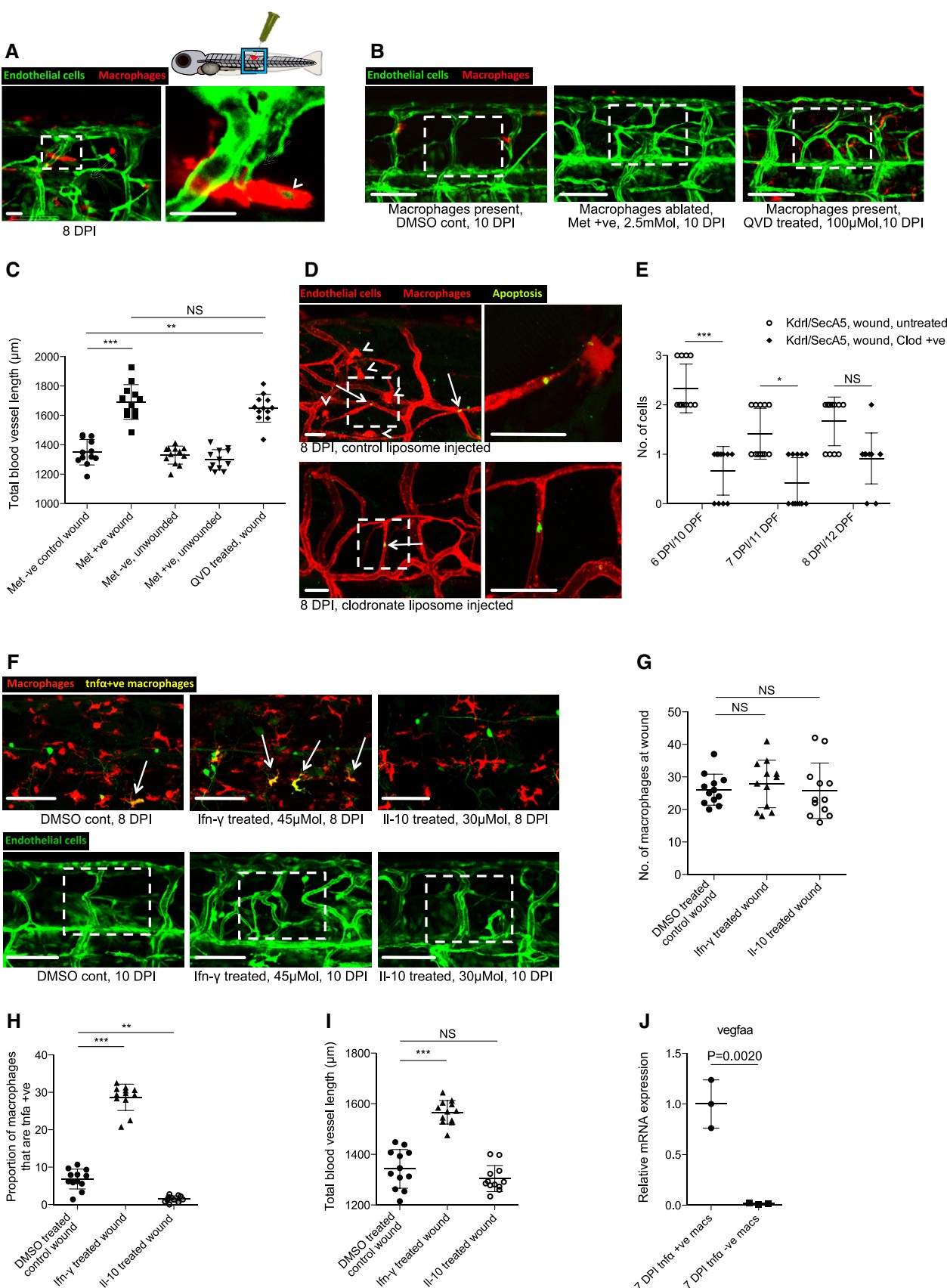

**Figure 7.**

Figure 7. **Macrophages mediate blood vessel regression, dependent on phenotypic status.**

A   Representative confocal projection images taken from needle-stick wounded Tg(*fli*:GFP); Tg(*mpeg*:mCherry), double-transgenic zebrafish, imaged at 7 DPI. Some macrophages at this later wound repair timepoint contain phagocytosed GFP tagged endothelial material (arrowhead). Boxed area denotes macrophage containing endothelial cell material. *N* = 9 independent fish.

B   Representative confocal projection images taken from needle-stick wounded Tg(*fli*:GFP); Tg(*mpeg*:KalTA4); Tg(*UAS*:nfsB-mCherry) triple transgenic zebrafish, imaged at 10 DPI, treated with vehicle control, metronidazole or QVD pan-caspase inhibitor from 6 DPI. Both macrophage ablation and caspase blockade resulted in a failure to remodel vessels. Boxed area denotes wound site.

C   Quantification of total blood vessel length, measured from images represented in (B). *N* = 12 independent fish per timepoint per condition. Statistical significance, as determined by one-way ANOVA, is *P* ≤ 0.0001.

D   Representative confocal projection images taken from needle-stick wounded Tg(*kdrl*:mCherry-CAAX); Tg(*secA5*-YFP); Tg(*mpeg*:mCherry) transgenic zebrafish, imaged at 8 DPI, injected with control liposomes or clodronate liposomes as indicated. Macrophages are observed within the wound site in control liposome-treated fish (arrowheads). The apoptotic marker is seen in thin, unicellular tubes during the regression phase (arrows). Ablation of macrophages during the regression phase results in decreased levels of vessel apoptosis.

E   Quantification of apoptotic endothelial cells during vessel regression, measured from images represented in (D). *N* = 12 independent fish per timepoint per condition. Statistical significance, as determined by one-way ANOVA, is *P* ≤ 0.0001.

F   Representative confocal projection images taken from either 8 DPI Tg(*mpeg*:mCherry); Tg(*tnfα*:GFP) transgenic fish or 10 DPI Tg(*fli*:GFP) transgenic fish, treated as indicated from 6 DPI, following needle-stick injury. Inflammatory *tnfα*-positive macrophages are indicated (arrows). Boxed area denotes wound site.

G   Quantification of number of macrophages at the wound site at 8 DPI, measured from images represented in (F). *N* = 12 independent fish per timepoint per condition. Statistical significance, as determined by one-way ANOVA, is *P* = 0.7412 (not significant).

H   Quantification of proportion of macrophages that are *tnfα*-positive inflammatory macrophages at the wound site at 8 DPI, measured from images represented in (F). *N* = 12 independent fish per timepoint per condition. Statistical significance, as determined by one-way ANOVA, is *P* ≤ 0.0001.

I   Quantification of total blood vessel length at 10 DPI, measured from images represented in (F) using Angioanalyser. *N* = 12 independent fish per timepoint per condition. Statistical significance, as determined by one-way ANOVA, is *P* ≤ 0.0001.

J   Gene expression for FAC-sorted Tg(*mpeg*:mCherry); Tg(*tnfα*:GFP) zebrafish wound macrophages extracted at 7 DPI following needle-stick injury. *tnfα*-positive inflammatory macrophages express higher levels of *vegfaa* than *tnfα*-negative non-inflammatory macrophages. *N* = 100 independent fish per condition, 3 replicates. Statistical significance is indicated, as determined by two-tailed *t*-test.

Data information: For each ANOVA test, subsequent Bonferonni multiple comparison test was performed to determine level of significance, as indicated. Significance values: *$P$ ≤ 0.05, **$P$ ≤ 0.001, ***$P$ ≤ 0.0001. For all graphs, error bars indicate mean ± SD. Scale bars: (A, D) 20 μm; (B, F) 100 μm.

altering the total number of macrophages at this timepoint (Fig 7F–H). The same treatment regime in Tg(*fli*:GFP) transgenic fish leads to a clear failure in vessel regression at the wound site (Fig 7F and I). Finally, we performed quantitative PCR for *vegfaa* on FAC-sorted *tnfα*-positive versus *tnfα*-negative macrophages from these later stage (7 DPI) zebrafish wounds. Our results confirm that the inflammatory macrophages express higher levels of *vegfaa* than non-inflammatory macrophages at these later wounds, a scenario similar to that observed in early wound repair (Fig 7J). Taken together, these results demonstrate that appropriate vessel resolution is dependent not only on the presence of macrophages, but that these macrophages must be of a particular phenotype that no longer expresses pro-angiogenic signals.

# Discussion

In this study, we describe the process of wound angiogenesis and its resolution in mouse and zebrafish and investigate how the innate immune cell influx into the wound impacts on these neoangiogenesis events. We show that neutrophils are only transiently drawn to the tips of damaged vessels whereas macrophages exhibit long-term, intimate associations throughout the duration of vessel sprouting and anastomosis; at the later resolution stages, these wound macrophages contain phagocytosed vessel material. Macrophages are critically important for neoangiogenesis in wound repair, and ablating macrophages at either early or late stages of wound healing disrupts vessel sprouting and vessel regression. The early influx of macrophages is largely pro-inflammatory and express *tnfα*, but the proportion of these macrophages diminishes over time. By modulating this phenotypic switch with exogenous chemokines or genetically, we can skew macrophages towards a pro- or anti-inflammatory state at

early or late stages in the zebrafish wound repair process; these *in vivo* experiments suggest that pro-inflammatory macrophages express *vegfaa* and are required for early vessel sprouting, but that *tnfα* must be switched off for macrophages to effectively participate in vessel clearance. Our *in vitro* co-culture experiments confirm these interactions and reveal that *VEGFA* expression is a key component of the pro-sprouting capacity of inflammatory macrophages that preferentially associate with sprouting tips.

A significant and unique advantage of performing our live-imaging studies in zebrafish larvae is the ease of imaging in translucent tissues. Unlike the opaque tissues of mouse that require complicated and invasive procedures to image, we can visualise blood vessels prior to injury induction and can therefore selectively injure areas of living zebrafish larval tissue with or without damage to adjacent vessels. This has enabled us to observe that wound angiogenesis occurs in response to tissue damage regardless of whether there is damage to existing vessels. Another outcome of our live-imaging studies is that wound neoangiogenesis appears to largely recapitulate earlier episodes of embryonic developmental angiogenesis. The resultant intimate macrophage–vessel association shown in our wounding models appears to mimic certain elements of what has previously been observed in developmental contexts such as retinal (Rymo *et al*, 2011) and hindbrain angiogenesis in embryonic mice (Fantin *et al*, 2010), and in the connection of intersomitic vessels in the embryonic zebrafish trunk (Fantin *et al*, 2010). Indeed, the dynamic nature of filopodial formation and vessel sprouting during repair of larval zebrafish wounds appears to replicate that observed during normal developmental angiogenesis (Isogai *et al*, 2003).

A recent study in which zebrafish vessels were damaged by hypoxia also reports that the responding macrophages are largely *tnfα* positive (Gerri *et al*, 2017). Our live-imaging studies in fish combined with *in vitro* studies go beyond this and demonstrate that

this inflammatory phenotype is needed for sprouting in wounds and that the pro-inflammatory macrophage-mediated sprouting is facilitated by *vegf*. Clearly, the *VEGF* signal does not have to be locally delivered because macrophage-conditioned medium added to our *in vitro* co-culture vessel assay triggers increased vessel sprouting and outgrowth. However, macrophages, particularly inflammatory ones, associate with vessel tips in both human co-cultures and fish wounds, suggesting that macrophages will tend to act as point sources of *vegf* signalling. Similar, *vegf*-based macrophage–endothelial cell guidance has previously been suggested in the context of mediating repair of peripheral nerves by bridging the gap between damaged nerve ends (Cattin *et al*, 2015); our studies reveal that this mechanism is part of the inflammatory response to injury which must be tightly controlled in order to establish fully functional, patent vessels in a variety of wound repair scenarios.

During the later resolution stages, we observe that macrophages contain phagocytosed vessel material. Furthermore, ablation of macrophages during later phases of repair in fish wounds results in a failure to appropriately remodel the vasculature, and this failure is reproduced by global inhibition of apoptosis. The dynamics of vessel pruning and regression in our wounds also appears to be very similar to that observed in zebrafish retinas during embryo development, with apoptotic endothelial cells observed almost exclusively in thin, unicellular tubes as blood vessels progress from multicellular lumenised structures to the point of detachment (Kochhan *et al*, 2013). Whether wound macrophages simply clear away this vessel death, or are actively driving vessel remodelling by directing endothelial cell death as previously reported in hyaloid vessel regression in mouse retina (Lang & Bishop, 1993; Lobov *et al*, 2005; Rao *et al*, 2007), is an area of active investigation. We observe that appropriate vessel regression fails if pro-inflammatory signals persist or are re-stimulated at later stages of repair, and we show that one of the mechanisms preventing this regression is continued *vegf* expression by inflammatory macrophages. This experimental scenario may reflect pathologies in which the dysregulation of wound angiogenesis is associated with an altered inflammatory profile, such as atherosclerosis or chronic wounds (Frykberg & Banks, 2015; Malecic & Young, 2017). Understanding how and when to nudge the innate immune wound response towards a pro-inflammatory state to stimulate sprouting and then away from this state in a timely fashion to allow for vessel regression is therefore critical in devising new therapies to deal with a broad range of pathologies.

It is interesting that during normal wound repair we see neutrophils displaying very transient associations with vessel tips, in contrast to the much longer macrophage–vessel interactions. These wound neutrophils express soluble vegf receptor 1 (*sflt1*), which means they may act as an angiogenic brake during the early period following wounding; indeed, when we delete macrophages, we see that neutrophils persist at these sites for considerably longer than when macrophages are present, and, as a consequence, wound neoangiogenesis fails. This presence of anti-angiogenic neutrophils could explain previous observations of increased and prolonged neutrophil numbers in human chronic wounds, which fail to heal in part due to lack of appropriate revascularisation (Martin & Nunan, 2015). Furthermore, early wound macrophages express genes involved in mediating neutrophil resolution from wounds, a function that has previously been demonstrated in other wound contexts (Tauzin *et al*, 2014; de Oliveira *et al*, 2016). These results indicate

that as well as their direct roles in regulating wound angiogenesis, macrophages play an indirect role in controlling when the brakes to angiogenesis are removed by inducing anti-angiogenic neutrophils to depart the repairing wound. How neutrophils and macrophages identify damaged vessel tips that they subsequently associate with, and whether macrophages play further roles in regulating the behaviours of other important cells in the repair process that may influence endothelial cells—such as fibroblasts and pericytes—remain open questions.

In larger wounds, we see numerous macrophages associating with damaged vessels and driving vessel sprouting, whereas in smaller vessel laser lesions we observe individual macrophages straddling the gap between two vessel tips, just as has also been shown in a recent zebrafish stroke model of brain vessels microlesions (Liu *et al*, 2016). We presume that macrophages facilitate the repair of vessels in larger injuries in a stepwise manner, firstly encouraging sprouting angiogenesis and extension of vessel tips via vegf signalling until they reach close proximity with one another and then, just as in repair of microlesions, support ligation and anastomosis at sites of vessel fusion. Furthermore, we observe *tnfα* expression by the majority of early wound responding macrophages associated with sprouting tips, while later macrophages appear to downregulate *tnfα* expression during the anastomosis and vessel maturation process. Indeed, some of our timelapse experiments directly demonstrate *tnfα*-positive macrophages switching to a *tnfα*-negative phenotype, indicating a pro- to anti-inflammatory switch in macrophage phenotype previously observed only indirectly in a variety of mouse wound contexts (Arnold *et al*, 2007; Ramachandran *et al*, 2012). These *in vivo* data, along with our *in vitro* results showing that anti-inflammatory IL-4-treated macrophages lead to fewer blood vessels, suggest that differential action in macrophage function may be driven by macrophage phenotype switching. Defining precisely what mechanisms are utilised by macrophages to facilitate vessel sprouting, stabilisation and "nurturing," in addition to delivery of angiogenic factors, warrants further study. Moreover, whether the phenotype switching that we observe is normally triggered by exogenous factors or is a passive time-limited occurrence, and whether the manipulation of these phenotypic states will yield therapeutic benefits are all important questions for further investigation.

## Materials and Methods

### Mouse housing and excisional wounding

All experiments were conducted with approval from the local ethical review committee at the University of Bristol and in accordance with the UK Home Office regulations (Guidance on the Operation of Animals, Scientific Procedures Act, 1986). Male wild-type C57Bl/6J mice (7–10 weeks old) were purchased from Charles River and kept under standard pathogen-free conditions with food (regular chow diet) and water *ad libitum* and 12:12-h light–dark cycle. For skin wounding experiments, mice were anaesthetised with isoflurane and four full-thickness excisional wounds made to the shaved dorsal skin (4-mm biopsy punch, Kai Industries). Wounds and 0.5 cm surrounding skin were harvested on days 0, 1, 3, 7, 10 and 14 after wounding and either prepared for cryosectioning or fixed overnight

in 4% paraformaldehyde at 4°C overnight on a rocker for whole-mount multiphoton imaging.

### Tissue preparation for multiphoton imaging

After fixation in 4% PFA, wounds were washed with PBS. Blocking and permeabilising was achieved with 1% Triton X-100, 1% BSA, 3% normal goat serum in PBS at room temperature overnight. Tissues were then exposed to antibodies diluted in block/perm solution incubated for 48–72 h at 4°C on a rocker. Thereafter, tissues were mounted in Prolong Diamond (Life Technology) for multiphoton imaging. For frozen sections, mouse wounds were fresh snap-frozen in optimal cutting temperature (OCT) compound (Tissue-Tek) on liquid nitrogen-cooled isopentane and sectioned at 50 μm.

### Multiphoton imaging

Two-photon imaging with z-compensation was performed to detect the fluorophore staining. Blood vessel endothelial cells were labelled with Alexa Fluor 594-conjugated rat anti-mouse CD31 (1 μg/ml, clone MEC13:3; Biolegend) (excited @ 760 nm) or Alexa Fluor 647-conjugated rat anti-mouse CD144 (2.5 μg/ml, clone BV13, Biolegend), and macrophages were labelled with Alexa Fluor 488-conjugated anti-mouse CD68 (2.5 μg/ml, clone FA-11; Biolegend—excited @ 710 nm). Multiphoton images were generated on a Leica SP8 MP/CLSM system (Leica Microsystems).

### Transmission electron microscopy

Wounds were harvested at the indicated timepoints, fixed and processed as previously described (Nunan *et al*, 2015). Ultrathin (0.02 μm) sections were images on a Tecnai 12-FEI 120 kV BioTwin Spirit transmission electron microscope.

### Zebrafish strains and maintenance

All experiments were conducted with approval from the local ethical review committee at the University of Bristol and in accordance with the UK Home Office regulations (Guidance on the Operation of Animals, Scientific Procedures Act, 1986). Wild-type and transgenic lines Tg(*fli1*:eGFP) [referred to as Tg(*fli*:GFP)] (Lawson & Weinstein, 2002), Tg(*mpx*:GFP)(Renshaw *et al*, 2006), Tg(*mpeg1*: mCherry) [referred to as Tg(*mpeg*:mCherry)] (Ellett *et al*, 2011), Tg(*mpeg1.1*:Gal4FF) [referred to as Tg(*mpeg*:Gal4)] (Palha *et al*, 2013), Tg(*UAS-E1b*:NfsB-mCherry) [referred to as Tg(*UAS*:NfsB-mCherry)] (Davison *et al*, 2007), TgBAC(*tnfα*:GFP)[referred to as Tg(*tnfα*:GFP)] (Marjoram *et al*, 2015), Tg(*dll4*:EGFP) (Sacilotto *et al*, 2013), Tg(*kdrl*:mCherry-CAAX) (Fujita *et al*, 2011), TgBAC(*etv2*: EGFP) [referred to as Tg(*etv2*:GFP)] (Proulx *et al*, 2010), and Tg(*tbp*):GAL4) together with Tg(*UAS*:SEC-Hsa.ANXA5-YFP,*myl7*:RFP) [referred to as Tg(*secA5*:YFP)] (van Ham *et al*, 2010) were maintained on TL wild-type background, and staging and husbandry were performed as previously described (Westerfield, 1995). Mutant strains used were *csf1ra^{j4e1}* (Parichy *et al*, 2000) and *vegfaa^{bns1}* (Rossi *et al*, 2016), maintained on AB background or used in combination with transgenic lines as indicated. *csf1ra^{j4e1}* mutants were genotyped by visual inspection for absence of mature xanthophores as previously described (Parichy *et al*, 2000). *vegfaa^{bns1}* mutants

were genotyped as previously described (Rossi *et al*, 2016), using primers 5′-CGAGAGCTGCTGGTAGACATC-3′ and 5′-GGATGTACG TGTGCTCGATCT-3′. *vegfaa^{bns1}* mutants were rescued by injection of 200 pg vegfaa-165 RNA into the cell at the one-cell stage, as previously described (Rossi *et al*, 2016).

### Needle-stick injury

Needle-stick wound induction was performed into the dorsal somites opposite the cloaca with either a 30-gauge needle (Becton Dickinson) or fine tungsten needle (Harvard Apparatus), as previously described (Gurevich *et al*, 2016). Subsequently, fish were immediately transferred to fresh embryo water (E3) to recover. Sham wounded fish were used as an uninjured control. Sham and wounded fish were fed as normal from 6 days post-fertilisation, and fish were rejected from analysis for experiments dealing with vessel regression if they had not grown to at least 5 mm in length by 10 DPI, as this was taken as a sign of insufficient feeding. Fluorescent stereomicroscope images were generated on a Leica M205 FA system (Leica Microsystems). Confocal images were generated on a Leica SP8 MP/CLSM system (Leica Microsystems).

### Laser injury

Laser-induced injury was performed as previously described (Gurevich *et al*, 2016), with some modifications. Briefly, anaesthetised fish were mounted directly into a glass-bottomed 35-mm dish (Mattek). A Micropoint laser (Spectra Physics) connected to a Zeiss Axioplan II microscope (Zeiss Microimaging) was used for injuring as previously described (Otten & Abdelilah-Seyfried, 2013), using a 40× water immersion objective and a laser pulse at a wavelength of 435 nm for cell ablation. Timelapse recordings were generated on a Leica SP8 MP/CLSM system (Leica Microsystems).

### Fluorescent angiography

Zebrafish fluorescent angiographies were performed as previously described (van Rooijen *et al*, 2010), with some modifications. Briefly, high molecular weight rhodamine–dextran conjugate (2,000,000 MW, Sigma) was injected into the caudal vein of fish prior to imaging at the timepoints indicated.

### Macrophage ablation experiments

Metronidazole cell ablation was performed as previously described (Pisharath & Parsons, 2009; Gurevich *et al*, 2016), with some modifications. Briefly, the pro-drug metronidazole (Sigma) was dissolved in E3 water as a 10 mM or 2.5 mM working solution, as indicated, along with added 0.2% dimethyl sulfoxide (DMSO). Transgenic zebrafish were incubated at 28.5°C in 10 mM solution from 1.5 DPF to 4 DPF and 2.5 mM immediately after injury (4 DPF) as indicated, with solutions changed every day. Control fish were subjected to the same injury and treatment procedure without metronidazole.

Clodronate liposome (Clophosome A, FormuMax) cell ablation was performed as previously described (Bernut *et al*, 2014; Carrillo *et al*, 2016), with 5 nl of either Clodronate or PBS containing liposomes injected into the caudal vein of 3 DPF fish prior to injury at 4 DPF (10-nl injections used for 6 DPI fish).

## Drug and recombinant protein treatments

Fish were treated with compounds as described. 0.2% DMSO was used for all treatments as well as vehicle control. Lipopolysaccharide (referred to as LPS) (Sigma-Aldrich) was dissolved in PBS to 5 mg/ml as a stock solution and used as a working solution of 100 µg/ml diluted in E3 water. Hydrocortisone (Sigma-Aldrich) was dissolved in DMSO as a 500 mM stock solution and used as a working solution of 100 µM diluted in E3 water. Recombinant zebrafish Ifn-γ, IL-10 and IL-4 (Kingfisher Biotech) were all dissolved and maintained as a stock solution and used at working concentrations as indicated, diluted in E3 water. SKLB1002 and SU5416 Vegfr inhibitors (both Sigma-Aldrich) were dissolved in DMSO and used at working concentrations of 2.5 and 5 µM, respectively. Q-VD-OPH pan-caspase inhibitor (referred to as QVD) (Sigma-Aldrich) was dissolved in PBS to 100 mM as a stock solution and used as a working concentration of 100 µM. Fish were treated in all reagents at 28.5°C at the timepoints indicated with solutions changed every day. Control fish were subjected to the same injury and treatment procedure without compound exposure.

## Whole-mount *In situ* hybridisation

*In situ* hybridisation was performed using a commercial vegfaa probe (Dr-vegfaa; catalog no. 543241) and RNAscope® technology (ACD, USA) on 4% PFA fixed larvae following manufacturers protocols.

## Peripheral blood apheresis cones

Peripheral blood apheresis cones were acquired for research use in accordance with the Declaration of Helsinki and approval from the Bristol Research Ethics Committee (REC 12/SW/0199). Blood from apheresis cones was diluted 1:1 with Hanks balanced salt solution (HBSS, Sigma) containing 0.6% acid citrate dextrose (ACD) and layered onto Histopaque (Sigma, ρ1.077). Samples were centrifuged at 400 g for 30 min without brakes to generate a density gradient.

## Native neutrophil isolation

The interphase, plasma and histopaque layers were removed from the density gradient leaving only the red cell and neutrophil pellet at the bottom. The red cells were lysed in ice-cold lysis buffer (155 mM NH4Cl, 0.137 mM EDTA, 1 mM KHCO3, pH 7.5), on ice for 10 min. Centrifugation steps were carried out at 400 g at 4°C; lysis and centrifugation steps were repeated as necessary. Neutrophils were washed twice in cold PBS + 0.5% human serum albumin (HSA) before being resuspended in incubation media (RPMI + 20% FBS + 1% pen/strep), with or without 20 ng/ml Interferon α (IFN-α, Biolegend).

## Macrophage culture

Human peripheral blood mononuclear cells (PBMNCs) were isolated as previously described using the interphase layer of the density gradient (Ramos *et al*, 2013). CD14$^+$ cells were isolated from PBMNCs using a magnetic bead CD14$^+$ kit (Miltenyi Biotec), LS columns (Miltenyi Biotec) and MidiMACS™separator (Miltenyi Biotec) as per the manufacturer's instructions. CD14$^+$ cells were frozen in 50% foetal calf serum (FCS, Gibco), 40% PBS with 10% DMSO (Sigma) and stored in liquid nitrogen until required. Thawed cells were resuspended at a density of $0.33 \times 10^6$/ml in RPMI 1640 (Gibco) supplemented with 10% FBS (Gibco), 10 ng/ml macrophage–colony-stimulating factor (wt/vol, M-CSF, Miltenyi Biotec), and penicillin/streptomycin at 100 U/0.1 mg per ml of media, respectively (wt/vol; Sigma); with or without the inclusion of interleukin-4 at 20 ng/ml (wt/vol, IL-4, Biolegend), interferon-γ at 2.5 ng/ml (wt/vol, IFN-γ, Biolegend), for directed differentiation into pro- and anti-inflammatory macrophages. Cells were incubated at 37°C with 5% CO$_2$. Full media changes were performed twice throughout the 7-day culture where any cells in suspension were collection via centrifugation at 300 g and replaced in the culture. Cells were harvested from the adherent macrophage culture using a detaching buffer (10 mM EDTA, 15 mM Lidocaine in PBS) for inclusion in the co-culture assay.

## HUVEC co-culture

Normal human adult-derived fibroblasts (NDHF, Lonza CC-2511) were cultured in 12-well plates in HUVEC medium (EBM-2, Lonza), seeded at $3 \times 10^4$ cells/ml onto glass coverslips triple coated with 50 µg/ml fibronectin (F0895, Sigma-Aldrich), 30 µg/ml type I collagen (5409, Advanced BioMatrix) and 0.1% gelatin (G1393, Sigma-

**Table 1. Primers used for QPCR and genotyping.**

| Gene name | Forward primer 5′–3′ | Reverse primer 5′–3′ |
|---|---|---|
| ef1a (danio) | CTTCTCAGGCTGACTGTGC | CCGCTAGCATTACCCTCC |
| soluble flt1 (danio) | GCAAAAGGAGCGCAAACAGA | TTGGCGTCTGGGTTAGTGAC |
| membrane flt1 (danio) | AGCCCTGGGAAGATCAAAGC | GCTCAGACGAGGCTGACTTT |
| yrk (danio) | CAGATCATGAAGAGGCTCCGTC | CCTGCTCCAGCACCTCTCGG |
| cyba1 (danio) | ATGGCGAAGATTGAGTGGGCGAT | GCTGCAGCATGGAGGTTATCTGCT |
| vegfaa (danio) | AAAAGAGTGCGTGCAAGACC | AGCACCTCCATAGTGACGTT |
| vegfaa (danio, genotyping) | CGAGAGCTGCTGGTAGACATC | GGATGTACGTGTGCTCGATCT |
| EF1a (homo) | TGTCGTCATTGGACACGTAGA | ACGCTCAGCTTTCAGTTTATCC |
| TNFa (homo) | CGCTCCCCAAGAAGACAG | AGAGGCTGAGGAACAAGCAC |
| MRC1 (homo) | TTCAGAAGGTTTTACTTGGAGTGA | TCTCCATAAGCCCAGTTTTCA |
| VEGFA (homo) | AGGGCAGAATCATCACGAAGT | AGGGTCTCGATTGGATGGC |

Aldrich) in PBS, incubated for 4 days at 37°C/5% $CO_2$. Media was refreshed every 2 days throughout the assay. HUVECs (Lonza pooled human donors, CC-2519) were subsequently added to confluent fibroblast monolayer on day 5 at $3 \times 10^4$ cells/ml, cultured in HUVEC medium (Lonza EBM-2), incubated for a further 3 days at 37°C/5% $CO_2$. At this point, macrophages at $1.5 \times 10^5$, conditioned media or RPMI control media, were added to the co-culture assay, as indicated, and incubated for a further 3 days at 37°C/5% $CO_2$. Coverslips were subsequently fixed with 4% paraformaldehyde and immunofluorescently stained as previously described (Richards & Mellor, 2016), using mouse monoclonal anti-human CD31 antibody (1:200 of 0.5 mg/ml stock, BBA7, R&D Systems) to label endothelial cells, mouse monoclonal anti-human MCSFR antibody (1:200, ab183316, Abcam) to label macrophages, DAPI (D21490, Invitrogen) and goat anti-mouse secondary antibodies Alexa Fluor 488 (1:400, A11029, Invitrogen) and Alexa Fluor 594 (1:400, A11032, Invitrogen). Human $VEFG_{165}$ antibody (AF-293, R&D Systems, referred to as VEGF blocking antibody) was used as a neutralising antibody at a working concentration of 200 ng/ml, incubated from the moment of macrophage/conditioned media addition and replaced with fresh VEGF antibody at each media change.

### Analysis of gene expression

For analysis of whole wounds, injured fish were anaesthetised, placed on a $3' \times 1'$ glass slide in a drop of water, manoeuvred into a lateral lying position and wounds were dissected using a surgical scalpel. Uninjured fish were similarly processed, with control unwounded tissue collected from the dorsal somites opposite the cloaca. Collected tissue was processed and RNA extracted using a RNeasy Mini Plus Kit (Qiagen) as per manufacturer's instructions. For analysis of sorted cells, wounds were harvested from transgenic zebrafish and dissociated to generate single-cell suspensions as previously described (Trinh *et al*, 2017). FACS analysis was performed using a Becton Dickinson InFlux (BD Biosciences San Jose, CA USA) and RNA extracted using a RNeasy Micro Kit (Qiagen) as per manufacturer's instructions. For analysis of cultured macrophages, cells were harvested and RNA extracted using a RNeasy Mini Plus Kit (Qiagen) as per manufacturer's instructions. cDNA was generated using a Maxima First Strand cDNA Synthesis Kit (Thermo Fisher) as per manufacturer's instructions. Quantitative PCR was performed using a SYBR Green PCR kit (Qiagen) and primers for genes as indicated (see Table 1) in an Agilent MX3005P QPCR cycler (La Jolla). Gene expression data were normalised against Elongation factor 1-alpha. 1% agarose gels were run at 80 V and stained with ethidium bromide at 0.5 μg/ml.

### Image analysis and statistics

All image analysis was performed in Velocity (Perkin Elmer) or ImageJ. Pixel counts were performed in ImageJ. Vessel orientation was measured using the ImageJ plugin OrientationJ, which rates the coherence of neighbouring pixels between 0 (isotropic, no overall direction) and 1 (one dominant direction; Fonck *et al*, 2009). Vessel metrics (such as total vessel length, node and sprout/branch numbers) were calculated by using the ImageJ macro, Angioanalyser (Gholobova *et al*, 2015). All statistical analysis was performed

using Graphpad Prism. Data were confirmed to be normally distributed via d'Agostino–Pearson test or Shapiro–Wilk test prior to further comparisons. Student's t-test was used except in the case of comparisons involving more than two groups; in these instances, one-way ANOVA was performed for all comparisons, and a Bonferroni multiple comparison test was subsequently performed.

**Expanded View** for this article is available online.

### Acknowledgements
We thank Deborah Carter for technical support with electron microscopy; Debra Ford for technical support with cryosectioning; Robert Nunan for mouse wound TEM samples; Jimmy Wei for assistance with HUVEC co-culture; and members of Rebecca Richardson's and PM's laboratories for helpful discussion. We also thank the team at the Wolfson Bioimaging Facility (Bristol, UK) for their help with imaging and image analysis, as well as Sarah De Val (Oxford University, UK) and Michele Marass and Didier Stainier (Max-Planck, Germany) for fish lines. DBG, JC and PM were funded by a Wellcome Trust Investigator Award (WT097791/Z11/Z). AG was funded by Cancer Research UK programme grant (C19/A11975). CT and HM were funded by a Wellcome Trust PhD studentship (102390/Z/13/Z). CES and AMT were funded by grants from NHS Blood and Transplant (WP15-05) and the National Institute for Health Research Blood and Transplant Unit (NIHR BTRU) in Red Blood Cell Products at University of Bristol in Partnership with NHS Blood and Transplant (NHSBT). NIHR did not fund the animal work described in this study. The views expressed are those of the authors and not necessarily those of the NHS, the NIHR, the Department of Health or NHSBT.

### Author contributions
DBG designed and performed zebrafish wounding, treatment and imaging experiments and mouse wound electron microscopy imaging; DBG, CES, CT, AMT and HM designed and performed human tissue co-culture experiments; AG designed and performed quantitative PCR experiments; JC designed and performed mouse wounding and multiphoton imaging experiments; PM supervised the project. DBG and PM wrote the manuscript.

### Conflict of interest
The authors declare that they have no conflict of interest.

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
