## [Review Process File · The EMBO Journal]

Live imaging of wound angiogenesis reveals macrophage orchestrated vessel sprouting and regression

David B. Gurevich, Charlotte E. Severn, Catherine Twomey, Alexander Greenhough, Jenna Cash, Ashley M. Toye, Harry Mellor and Paul Martin

Review timeline:

Submission date:	17 th July 2017
Editorial Decision:	23 rd August 2017
Revision received:	21 st February 2018
Editorial Decision:	28 th March 2018
Revision received:	25 th April 2018
Accepted:	30 th April 2018

Editor: Karin Dumstrei

Transaction Report:

1st Editorial Decision

23rd August 2017

Thanks for your response and for outlining how you can address the concerns raised.

Given this I would like to invite you to submit a suitably revised manuscript. I can extend the revision time to 6 months. You will get a chaser at 3 months and please let me know how everything is coming along at that stage.

Let me know if we need to discuss anything further

REFeree REPORTS

Referee #1:

The paper by Gurevich et al. investigates the influence of macrophages on the development of blood vessels following wounding. The authors use several animal models, such as mouse skin and embryonic zebrafish in addition to a HUVEC cell culture system to analyze the effects of manipulating macrophage numbers (in zebrafish) or their gene expression profiles on neo-angiogenesis. They define three different stages of macrophage involvement in mice: first macrophages are associated with tips of new blood vessel sprouts, macrophages subsequently "hug" new blood vessels and finally engulf material from dead endothelial cells. The next experiments are being conducted in zebrafish embryos, where the authors use either a needle punch or a laser to induce a wound. Here they first investigate blood vessel formation followed by an analysis of macrophage and neutrophil dynamics. They then study how an absence of macrophages influences neo-angiogenesis before manipulating the activation state of macrophages. In cell culture and zebrafish embryos, the authors finally provide evidence that macrophage-derived VEGF might be the pro-angiogenic stimulus regulating blood vessel growth after wounding. In general, the study

combines 3 different systems for studying the influence of macrophages on blood vessel growth. The mouse data (see below) and the cell culture data clearly need more in depth analysis. The more detailed analysis has been carried out in zebrafish, but it is not clear at present how the mouse and cell culture data would add to this. For the zebrafish part, the data are of high quality with adequate quantifications provided.

Major:

1. The authors claim that in mouse wound blood vessels become "more organized and returning to a distribution closely resembling unwounded skin by 14 DPI" (page 3). This statement cannot be supported by the panels provided in Figure 1B. The blood vessels at the 14 DPI time point do clearly look very different (e.g. in caliber and organization) from the unwounded condition. Evidently, a more detailed analysis on this part is necessary.

2. The authors use CD31 and CD68 to differentiate macrophages from endothelial cells and claim that in the clearance phase the staining of CD68 positive macrophages signifies the uptake of EC material. A recent publication shows that macrophages in skin lesions can express CD31 (Tidwell, Googe, Journal of Cutaneous Pathology, April 2014). The authors need to devise additional means to show an uptake of EC material by macrophages in order to substantiate their claim.

3. In order to show a conserved role of macrophages during neo-angiogenesis in mice, the authors need to deplete macrophages in mice, e.g. via clodronate treatment or use of CD11b-DTR mice, and analyse blood vessel development. Referring to my point 1, the mouse data lack sufficient depth of analysis at this point.

4. The authors are investigating different aspects of blood vessel formation after wounding in zebrafish. If I understand the paper correctly, they investigate for one the formation of new blood vessels in places, where there were no blood vessels before (needle damage) and on the other hand, they analyse the "repair" of blood vessels that were damaged in their normal place (microlaser). These two processes might rely on different mechanisms and need to be characterized in more detail. The authors state that "sprouting during wound angiogenesis appears to follow a similar process to that reported for embryonic development of the ISVs" (page5). A recent report (Wild et al., le Noble F., Nature Communications) showed that later sprouts mainly emanate from veins. Is this also true during wound healing? The authors state that for small ablations vessel outgrowth generally occurs in a ventral to dorsal direction. Is this true for both venous ISVs and arterial ISVs? This would be important for comparing blood vessel sprouting during embryogenesis with sprouting during wound healing.

5. The top 5 parameters in Figure 2K and Figure 3E (control situation) should represent similar processes. Why are there big differences, e.g. in neutrophil presence at vessel (yellow bar), blood vessel sprouting or re-establishment of flow? In addition, Figures 2K is redundant to Figure 3E. Figure 4I again shows a similar representation of the control situation.

6. The cell culture experiments (Figure 5) do not provide novel mechanistic insights. For instance, it is well established that IFN gamma stimulation of macrophages increases VEGF expression (and thereby angiogenesis), e.g. Ramanathan, Giladi, Leibovich, Experimental Biology and Medicine, 2003; Xiong, Elson, Legarda, Leibovich, Am. J. Pathol., 1998.

Referee #2:

In their manuscript "Live imaging of wound angiogenesis reveals how macrophage phenotypic state orchestrates vessels sprouting and subsequent regression during repair", the authors primarily utilized the zebrafish model to show how macrophages at different phenotypic states contribute to wound neoangiogenesis and vascular regression in a wound healing scenario. The optical transparency of zebrafish embryos was fully utilized to live image how macrophages at different phenotypic states physically interact with growing blood vessels during wound repair. Of particular importance, the authors suggest that macrophages have differing roles in angiogenic repair

depending on their inflammatory states. *tnf*-positive Inflammatory macrophages express *yrk* and *cyba1* proposed to repel anti-angiogenic *sflt1*-expressing neutrophils, and express *vegfaa* proposed to stimulate wound neoangiogenesis, while *tnf*-negative macrophages induce vascular apoptosis and phagocytose endothelial cells to promote vascular regression.

Overall, this research has revealed novel mechanisms for wound neoangiogenesis and vascular regression during wound healing that may have important clinical relevance. The data are mostly clear and convincing but there are a few points that need further evidence and areas that should be improved before publication.

Major comments:

- 1) In the zebrafish: the QPCR data suggesting that *Vegfaa* is responsible for regulating angiogenesis in wounds is supported by treatment with a single chemical inhibitor only. In the absence of a tissue specific loss of function study for macrophage *Vegfaa*, further cumulative evidence is needed to support that *Vegfaa* is responsible for wound neo-angiogenesis. It would be helpful to see the macrophage expression of *Vegfaa* in situ in the *TNF α* positive macrophages. It would also be necessary to show that multiple *Vegfr2* signalling inhibitors give the same effect to be confident of the specificity of that result. More convincing would be to perform the wounding assay in the *vegfaa* genetic mutants recently reported by Rossi et al (2016) which can be rescued for early developmental defects by simple mRNA injection and would thus serve as a workable model to test this hypothesis in a mutant setting.
- 2) Although it is clear that macrophages are required for vascular regression, the link between macrophages and vascular apoptosis is still weak. Please show whether *tnf*-negative macrophages at 10DPI express more macrophage paracrine factors that were shown to induce vascular apoptosis such as *wnt7b*, *ang2* or similar when compared to *tnf*-positive inflammatory macrophages. It would also strengthen claims to show whether macrophages isolated from unwounded fish express less of these paracrine factors (as macrophage driven vascular apoptosis may be less frequent in unwounded fish).
- 3) For all of the data presented, the use of bar graphs and the absence of individual datapoints in graphs weakens the accessibility of results. The paper would be improved by showing dot-plots throughout to make the individual *n*-values for each experiment more obvious.
- 4) Figure 1B: The authors have shown that wound neoangiogenesis in mice becomes more organised and returns to a distribution closely resembling unwounded skin by 14DPI. Although the quantification of blood vessel coverage indicates this, images in Figure 1B show that the blood endothelial cells in 14DPI mice are still disorganised. The manuscript would be improved by quantifying organisation of the vessels.
- 5) Figure 1D, 7A: How often do you see macrophages with engulfed endothelial cell material? This should be quantified in both mice and fish.
- 6) The fish were raised to 10 days post-injury (14 days post-fertilisation). Please describe whether fish were fed after 6-7 days post-fertilisation (when yolk sac is largely depleted). If yes, please include this in the method section. If not, some of the data may require reproduction as starvation state of the fish may significantly skew the results observed.
- 7) It is quite clear from the Movie 1 that neutrophils are only drawn to the vascular tip for a short period of time. Similar to the one done with macrophages, it would be helpful to quantify the percentage of sprout tips associated with neutrophils.
- 8) The *tnf*-positive Inflammatory macrophages had been shown to only interact with vascular tip cells during neoangiogenesis and at later stage (4DPI), this is replaced with *tnf*-negative macrophages. Please indicate whether the *tnf*-negative macrophages are *tnf*-positive Inflammatory macrophages that switched phenotype or whether fresh *tnf*-negative macrophages are recruited to the site of injury over time.

Minor comments:

- 1) It would be nice to see the correlation between macrophage number and wound recovery plotted as a correlation and supported by an r^2 value.
- 2) The QPCR mFlt1 negative data should be shown. Was there a positive control for that experiment?
- 3) Can you comment whether the repaired vasculature roughly resemble unwounded vasculature? If so, there may be guidance factors that direct neoangiogenesis along pre-determined routes.
- 4) Figure 2G: does the author mean percentage of sprout tips with associated macrophage?
- 5) Page 6: macrophages, "neutrophils are retained at the wound site and a maintain a long" should be "and maintain a long".
- 6) Similar to Tazuin et al., 2014 (reference 33), did you see contact-dependent neutrophil reverse migration in your wound neoangiogenesis model? Were these neutrophils associated with the vascular tip before making contact with macrophage?
- 7) Figure 5C: If proportions of macrophage were quantified, would the total number of proportions roughly add up to 10? Please clarify how these proportions were calculated.
- 8) Figure 7C: figure labels need fixing

Referee #3:

Gurevich et al. show that macrophages are essential for wound angiogenesis and associate with new sprouting blood vessels. They find that pro-inflammatory macrophages in particular play a role in initiating and driving vessel sprouting, at least in part via the release of VEGF-A. Additionally, macrophages also affected wound angiogenesis indirectly by dispersing anti-angiogenic neutrophils that are recruited in the early stages of wound repair. Finally, the authors demonstrate that macrophages undergo a switch to an anti-inflammatory phenotype during the resolution phase and mediate blood vessel regression.

Others have already reported the involvement of macrophages in developmental and physiological angiogenesis, as well as in wound angiogenesis. According to the authors, the dynamic nature, precise function of the macrophage phenotypes, exact timing of interaction and roles of other leukocytes such as neutrophils remained to be investigated. To address these questions, the authors mostly used real time in vivo imaging in zebrafish.

The time course of macrophage and neutrophil recruitment has already been studied in adult zebrafish tail fin regeneration (Petrie et al., *Development*, 2014). Also, the kinetics of macrophages recruitment after wounding and infection with *E. coli* in zebrafish has been previously reported by Nguyen-Chi et al. They showed that unpolarized macrophages are recruited to the site of inflammation and start expressing TNF- α . Afterwards they are converted into the M2 phenotype during the resolution phase (Nguyen-Chi et al., *eLife*, 2015). However, both publications did not study the effect on blood vessels. This is where the present manuscript provides novel and relevant insights.

Although most of the findings in this study are well supported by the data, existing issues with the used transgenic zebrafish lines lead to the fact that some of the conclusions are not very convincing. Previous experience by several labs with the Tg(*fli1a:eGFP*), the Tg(*fms:gal4-VP16*);Tg(UAS:*nfsB-mCherry*);Tg(*fli1a:eGFP*) and the Tg(*mpeg1:gal4*);Tg(UAS:*nfsB-mCherry*);Tg(*fli1a:eGFP*) lines, demonstrate that macrophages are sometimes labeled by the *fli* promoter and at the same time not labeled by the *fms* or *mpeg* promoter. (This can even be seen in some of the presented movies, see Movie 6.) For the Tg(*fli:GFP*);Tg(*mpx:GFP*) line with or without additional transgenes for macrophage labeling, this means that it is possible that the green labeled cells are not neutrophils,

but macrophages. On the other hand, the macrophages are sometimes also double labeled (green by the *fli* promoter in addition to the red signal by *fms* or *mpeg* promoter). For the *Tg(fli:GFP);Tg(mpeg:mCherry);Tg(tnf α :GFP)* line, this means that one would be led to assume that these macrophages are TNF- α positive, while this might not always be the case. For the *Tg(fli:GFP);Tg(mpeg:mCherry)* line, the authors cannot conclude with full confidence that the double labeled macrophages contain phagocytosed vessel material (although this may be likely, even if not mechanistically important). Additionally, it is important to note that there are often gene silencing issues with UAS transgenes. As a result the fraction of the macrophages labeled or not will vary considerably, in particular when breeder selection is not carefully managed. Secondly, the claim that neutrophils only appear early and transiently at the wound site, although interesting and potentially relevant, is not really supported by the data (see specific comments for Fig. 3c, Movie 1, and Movie 3).

Furthermore, the manuscript leaves a couple of questions unanswered. Do macrophages facilitate wound angiogenesis via other mechanisms besides VEGF signaling? The used methods to interfere with VEGF effects are not selective for cell-type derived VEGF and there is no direct evidence for macrophage derived VEGF being the main mechanism. Are macrophages really driving EC apoptosis during blood vessel regression or are they just clearing apoptotic ECs? What triggers their phenotype switch?

Curiously, the authors suggest that the pro-inflammatory macrophages are the major source of VEGF, whereas anti-inflammatory macrophages produce less. To this reviewer's knowledge, mouse studies rather suggest that the anti-inflammatory macrophages of the M2 polarization (characterized by MRC1 expression) are a major source of VEGF.

Overall, this work is very interesting, but requires some improvements concerning the issues with the transgenic lines. Providing more convincing movies (e.g. from other transgenic lines), will largely address the major concerns. However, there are other issues with the work that requires some attention, including statistics. I listed below a number of specific comments and additional non-essential suggestions for the authors will hopefully find useful for improving their interesting study.

SPECIFIC COMMENTS

Statistics

The statistical analysis presented is problematic. Normality of the data should be checked. If the data are not normally distributed, the authors need to transform them or use a non-parametric test. If the data are normally distributed and only have two independent groups, the t-test can be used. If there are more than 2 groups (which is the case for Fig. 3b; Fig. 4f; Fig. 5c,e,f; Fig. 7e,g,h,i; Sup. Fig. 2e, Sup. Fig. 3d and Sup. Fig. 4a), t-test is not acceptable. The authors should use ANOVA and correct for multiple testing to reduce the number of false positives.

Figures

Figure scale bars: sometimes the scale bar is present on every picture, sometimes only on 1 or 2 pictures or even missing completely. More consistency here would be useful.

Figure 1: (b) The picture of 14 DPI looks very different from the picture of the unwounded skin, although the authors describe in the text that it resembles unwounded skin. (c) Maybe quantifying other characteristics like vessel length and/or complexity like the authors did for fish later on would be useful, because now the quantification of blood vessel coverage for unwounded skin and wounded skin at 14 DPI is almost the same, but the pictures are not similar. The picture of 0 DPI does not seem representative for 20% blood vessel coverage. (d) Using a lower magnification would give a good overview of the different types of interaction. The authors could also stain for tip cells (e.g. ESM1) to really visualize that the macrophages are associated with tip cells in the early stage of wound angiogenesis. (e) The text is not really describing what can be seen from the graph. According to the text, the macrophages are associated with tip cells from 3-7 DPI and show a "hugging" behavior from 7-10 DPI. However, the percentage of macrophages associated with sprout tips at 7-10 DPI is not very different from 3-7 DPI.

Figure 2: (a) It should be "wounds" instead of "wouns" in the figure legend. (b) What do the different colors mean in the graphical representation of the nodes and sprouts? (d) The authors can

maybe indicate in the first picture that the fish is not wounded yet, and include a box that denotes the wound site. Maybe also include a picture of 12 HPI, since this is an important time point, namely the peak of neutrophil recruitment. (f) To make it more complete, the authors could also include pictures of neutrophil recruitment. (g) Label of the Y-axis of the right graph is incomplete. (i) Scale bar is missing at the left picture. It is not really clear that the vessels have anastomosed and are perfused at 930 MPI. It seems that this is happening outside the range of the z stack. Therefore, one cannot conclude from these pictures (and movie) that the sprouting only happens in a ventral to dorsal direction. In addition, the authors might want to use a transgenic line in which the neutrophils have a different color, for example labeled with CFP, because with this transgenic line, one cannot be certain that the green labeled cells are neutrophils. If the authors would use a third color, they can easily distinguish between ECs, macrophages and neutrophils. Moreover, it would be easier to differentiate ECs from neutrophils that are localized around blood vessels. (k) Similar comment as for (i), from these pictures and movies, one cannot conclude when anastomosis and perfusion are precisely happening.

Figure 3: (b) Please make the bar labels less confusing, maybe by mentioning (part of) the transgenic line that was used. Please be consistent in the colors of the bars: keep the same color for "Met treated ctrl injury" in Sup. Fig. 2e and Fig. 3b. (c) Same comment as for Fig. 2i about the transgenic line. One can see neutrophils here in the control embryo at 230 MPI, which is contradictory with the statement that neutrophils are only present from 30 to 90 MPI. (d) It would be useful to try to keep the labeling consistent, rather use "Untreated injury" and "Macrophage Met ablated injury" like in b instead of "Untreated control" and "Metronidazole treated injury", and try to use the same bar colors as in b. (f&g) Rather use "Wounded" instead of "Wound" in the bar labeling.

Figure 4: (b) The first picture is not representative for 50% of TNF- α positive macrophages like described in the text and visible in e. (In e the number is even higher than 50% for DMSO ctrl.) The third picture is also not representative for the percentage in e. (d&e) Include "WT" in the graph legend, like in Sup. Fig. 3b&c. (g) Same comment as for Fig. 2i: it would be more ideal to use a different color for TNF- α signal. In Tg(fms:gal4-VP16);Tg(UAS:nfsB-mCherry);Tg(fli1a:eGFP) and Tg(mpeg1:gal4);Tg(UAS:nfsB-mCherry);Tg(fli1a:eGFP) lines, the macrophages are sometimes double labeled (green by the fli promotor in addition to the red signal by fms or mpeg promotor). In this case, one would assume that the macrophages are TNF- α positive, while this might not always be the case. In the picture of 30 MPI, the arrow on the right seems to indicate a TNF- α negative macrophage instead of a TNF- α positive one. From these pictures, it is not really clear that TNF- α positive macrophages are interacting earlier and more frequently with sprouting vessels than TNF- α negative ones.

Figure 5: (a&b) What are the small red dots? Did the staining not work properly? DAPI labels all nuclei, not only fibroblasts? Why are there so few macrophages present? (e) Description of the second and third graph is missing in the figure legend.

Figure 6: (a) Description of second and third graph is missing in the figure legend. (c) Mistake in the figure legend, it should be: "..., measured from images represented in B." (d) Should it not be 4 DPF instead of 4 DPI above the pictures?

Figure 7: (a) Same comment about macrophage labeling by the fli promotor. It might be that the macrophages are green, just because they are labeled by the fli promotor, and not because they contain phagocytosed EC material. (b) Please be consistent in the name of the transgenic line. In the text you use "Tg(flkl:DsRed)" and in the figure legend you use "Tg(kdrl:DsRed)". Please indicate more clearly whether the picture came from a wounded or unwounded embryo. (c) Part of the labeling at the x axis is missing. (d) Again, please be more consistent in the names of the transgenic line. (g) Part of the y axis label is missing. Indication of statistical significance is missing.

Supplemental figure 1: Scale bars are missing. (a) inconsistent panel labeling: the authors could use "0 HPI" and "1 HPI" instead of "immediately post stab injury" and "one hour post stab injury". (b&d) Please include a box that denotes wound site. (d) Same comment as for figure 2i: the authors might want to use another color for the neutrophils, especially in this double transgenic line, since one cannot know whether green labeled cells are neutrophils or macrophages labeled by the fli promotor.

Supplemental figure 2: (a) Please be consistent in the names of the transgenic line, use either Gal4FF or KalTA4. (e) At what time point is the total vessel length quantified? This is not clearly indicated in the figure or figure legend. The authors could maybe indicate more clearly that "Met treated control injury" are Tg(fli:GFP) embryos treated with metronidazole.

Supplemental figure 3: (a) Please indicate what the colors represent (macrophages, TNF- α positive macrophages, etc.), like done for other pictures. (d) Include "WT" in the bar labeling like in Fig. 4f. (e) Same comment as for a. Why not show the green signal here, like in the other Dextran pictures (e.g. Fig. 2j)?

Supplemental figure 4: /
Movies & tables

The names of the movies files are confusing. In addition, all movies were uploaded twice. Please also include scale bars in the movies.

Movie 1: Similar comment as for figure 2i: in this movie, the anastomosis and lumenization at 600 MPI as described in the text is not visible. It seems that this is happening outside the range of your z stack. Therefore, one cannot conclude from this movie that the sprouting only happens in a ventral to dorsal direction. What is the big red shape at the upper right? Unlike what is claimed in the text about neutrophils appearing early and transiently at the wound site, one can see several neutrophils interacting with the vessels around 720 MPI.

Movie 2: It would be interesting to show this in the triple transgenic line that was used in Movie 1. Again, the same comment as for Figure 2i and Supplemental figure 1d about an extra color.

Movie 3: Again, same comment as for movie 2: there are still neutrophils interacting with the sprouting vessel at 370 MPI.

Movie 4: /

Movie 5: In this movie, it is not really clear that TNF- α positive macrophages are interacting earlier and more frequently with sprouting vessels than TNF- α negative ones. Using a different color for the TNF- α positive macrophages might solve this problem.

Movie 6: /

Table 1: Table 1 was included 2 times.

NON-ESSENTIAL SUGGESTIONS

- M&M: "Sham wounded fish were used as an uninjured control." Please clarify what is meant with "Sham wounded", because the fish are uninjured?
- Second paragraph of the introduction: "Blood vessel regression during post-natal retinal development is also driven by macrophages, as their absence completely abolishes this process (8-10)." This statement is incorrect as the cited work refers to the specific case of regressing hyaloid vessels. There is no evidence for this process in development regression withing intrretinal vasculature.
- Typing/spelling errors:
 - o Timecourse vs time course
 - o One hour vs 1 hour
 - o Microlaser vs micro laser
 - o Quantitative PCR vs qRT-PCR
 - o Please use μm instead of um and μg instead of ug.
 - o In vitro and in vivo should be always in italics.
 - o Nomenclature of transgenic lines: a semicolon should be used between two transgenes instead of a comma and the name should in italics.
 - o Abbreviations: the first time, the full word used with the abbreviation between brackets, afterwards only the abbreviation should be used

POINT BY POINT REPOSE.

In this letter our responses/rebuttal notes are in bold.

Angio paper referees

Referee #1

The paper by Gurevich et al. investigates the influence of macrophages on the development of blood vessels following wounding. The authors use several animal models, such as mouse skin and embryonic zebrafish in addition to a HUVEC cell culture system to analyze the effects of manipulating macrophage numbers (in zebrafish) or their gene expression profiles on neo-angiogenesis. They define three different stages of macrophage involvement in mice: first macrophages are associated with tips of new blood vessel sprouts, macrophages subsequently "hug" new blood vessels and finally engulf material from dead endothelial cells. The next experiments are being conducted in zebrafish embryos, where the authors use either a needle punch or a laser to induce a wound. Here they first investigate blood vessel formation followed by an analysis of macrophage and neutrophil dynamics. They then study how an absence of macrophages influences neo-angiogenesis before manipulating the activation state of macrophages. In cell culture and zebrafish embryos, the authors finally provide evidence that macrophage-derived VEGF might be the pro-angiogenic stimulus regulating blood vessel growth after wounding. In general, the study combines 3 different systems for studying the influence of macrophages on blood vessel growth. The mouse data (see below) and the cell culture data clearly need more in depth analysis. The more detailed analysis has been carried out in zebrafish, but it is not clear at present how the mouse and cell culture data would add to this. For the zebrafish part, the data are of high quality with adequate quantifications provided.

Major:

1. The authors claim that in mouse wound blood vessels become "more organized and returning to a distribution closely resembling unwounded skin by 14 DPI" (page 3). This statement cannot be supported by the panels provided in Figure 1B. The blood vessels at the 14 DPI time point do clearly look very different (e.g. in caliber and organization) from the unwounded condition. Evidently, a more detailed analysis on this part is necessary.

We have now replaced the 14 DPI image with a more representative one (Fig 1B) and have quantified the angiogenic response and its resolution by analysis of vessel orientation (Fig 1C). At 14 days the vessels appear to resemble those seen in unwounded tissue in amount and organisation more closely than at early repair stages; however, they are not completely back to the unwounded pattern and we have softened our statement here (Page 3).

2. The authors use CD31 and CD68 to differentiate macrophages from endothelial cells and claim that in the clearance phase the staining of CD68 positive macrophages signifies the uptake of EC material. A recent publication shows that macrophages in skin lesions can express CD31 (Tidwell, Googe, Journal of Cutaneous Pathology, April 2014). The authors need to devise additional means to show an uptake of EC material by macrophages in order to substantiate their claim.

It is indeed the case that CD31 can stain macrophages as well as vessels in the manner indicated by the cited paper, but this doesn't seem to be the case for us. Only a subset of macrophages co-stain with CD31, and they do so in discrete domains within the cell (probably a phagocytic vacuole) – we have included a projection through the 'Clearance' macrophage in Figure 1E to demonstrate this (Movie 1). We have also included an expanded view (Sup. Fig 1A) showing how only a subset of macrophages (green) co-stain ex-vessel material (red). And to be sure, we have also supplemented these data with new frozen sections where we have stained vessels with VE-Cadherin antibody, a marker of endothelial cells that is not expressed by macrophages; again, we observe ex-vessel material in macrophages (Sup Fig 1B). Our new

TEM data (see later) is also supportive that at these later stages, assorted macrophages have engulfed cell debris near vessels.

3. In order to show a conserved role of macrophages during neo-angiogenesis in mice, the authors need to deplete macrophages in mice, e.g. via clodronate treatment or use of CD11b-DTR mice, and analyse blood vessel development. Referring to my point 1, the mouse data lack sufficient depth of analysis at this point.

The murine clodronate ablation experiments suggested here by Reviewer 1 have already been performed in two previous publications and shown to alter vessel patterning following wounding (Lucas et al, 2010, Willenborg et al, 2012), as we have mentioned in our manuscript (page 2). The objective of our study was to try to extend our mechanistic insight of this process by directly observing the cell interactions underlying inflammation mediated wound angiogenesis, by taking advantage of the genetic and imaging opportunities in zebrafish. The mouse data collected for our paper is primarily to descriptively expand our understanding of the vessel repair timecourse and to distinguish each of the various macrophage-vessel interactions seen throughout wound repair, rather than attempting to address any mechanistic questions. However, to improve the depth of analysis of the mouse data, we have now undertaken some TEM imaging of the various stages of wound repair (Fig 1E). This allows us to more subtly describe each of the macrophage/vessel interaction events we observe in mouse which we then follow up mechanistically in fish.

4. The authors are investigating different aspects of blood vessel formation after wounding in zebrafish. If I understand the paper correctly, they investigate for one the formation of new blood vessels in places, where there were no blood vessels before (needle damage) and on the other hand, they analyse the "repair" of blood vessels that were damaged in their normal place (microlaser). These two processes might rely on different mechanisms and need to be characterized in more detail. The authors state that "sprouting during wound angiogenesis appears to follow a similar process to that reported for embryonic development of the ISVs" (page5). A recent report (Wild et al., le Noble F., Nature Communications) showed that later sprouts mainly emanate from veins. Is this also true during wound healing? The authors state that for small ablations vessel outgrowth generally occurs in a ventral to dorsal direction. Is this true for both venous ISVs and arterial ISVs? This would be important for comparing blood vessel sprouting during embryogenesis with sprouting during wound healing.

We think we may not have explained our various lesions clearly enough. We have made two categories of mechanical lesions: large wounds (using a 30G needle) that damaged vessels, versus small wounds (using a fine tungsten needle) that damaged tissue without puncturing a vessel. Both of this class of mechanical lesion trigger vessel sprouting which shows how vessel damage per se is not required for wound angiogenesis. The second class of lesion was more precise, made with a microlaser, to an individual vessel, and even these were not all the same. We made some mini lesions/cuts, so we could watch the rapidity of vessel tip sprouting and fusion, and we made some larger laser lesions where an intersomitic vessel was completely vapourised so we could observe filopodial sensing as the vessel tips searched for one another from a distance. We have now more clearly spelled out the different wounding regimes in the text (pages 4 and 5).

The referee raises a very interesting question. Do the ingrowing vessels derive from vein or artery? Because of this query, we have now injured fish with fluorescently identifiable arteries versus veins (a gift from Sarah De Val, Oxford), and we see that sprouts generally emanate from whichever vessel is damaged/closest to the area of injury, and that both arteries and veins are capable of readily contributing to wound angiogenesis, often leading to chimeric vein-artery derived vessels in the wound (see new Fig 2B)

5. The top 5 parameters in Figure 2K and Figure 3E (control situation) should represent similar processes. Why are there big differences, e.g. in neutrophil presence at vessel (yellow bar), blood vessel sprouting or re-establishment of flow? In addition, Figures 2K is redundant to Figure 3E. Figure 4I again shows a similar representation of the control situation.

The differences are present in these metrics due to the variation in extent of lesion as discussed in response to comment 4 above, and as we now more fully indicate in the text (pages 4 and 5). Larger injuries elicit a larger inflammatory response, and take longer to sprout, anastomose and re-establish flow, and we have shown appropriate controls for each experiment. We have now more fully explained this in the text on page 5 and indicated which injury was performed in each experiment.

6. The cell culture experiments (Figure 5) do not provide novel mechanistic insights. For instance, it is well established that IFN gamma stimulation of macrophages increases VEGF expression (and thereby angiogenesis), e.g. Ramanathan, Giladi, Leibovich, *Experimental Biology and Medicine*, 2003; Xiong, Elson, Legarda, Leibovich, *Am. J. Pathol.*, 1998.

We think that our experiments do show additional insights; for example, our more extensive analysis has revealed novel observations regarding macrophage-vessel interactions (the increased localisation of macrophages to vessel tips after macrophages have been IFN gamma stimulated), and it is worth nothing also that our experiments utilise human primary cells (macrophages and endothelial cells), rather than murine cells as in the papers this referee cites. Reviewer 3 (see later) asks us for evidence that the vegf that drives wound angiogenesis is largely derived from macrophages and of course this is very difficult to address *in vivo* in zebrafish because of difficulties in lineage specific knockdown, but our reductionist, *in vitro* experiments, where blocking antibodies against vegf deplete the effects of macrophage conditioned media, go some way to addressing this. But your concerns led us to think of other novel insights that this co-culture approach might offer. We have now purified human neutrophils that we leave untreated or treat with a type 1 Interferon, IFN alpha (to skew neutrophils towards an inflammatory phenotype, Pylaeva, Lang and Jablonska, *Front. Immunol.*, 2016). When we apply these neutrophils to our *in vitro* angiogenesis assay, we find that neutrophils, and particularly inflammatory neutrophils, appear to suppress blood vessel growth (Fig 5I, J), just as indicated by our fish imaging approaches. This again demonstrates the power of such complementary studies, and to the best of our knowledge is a novel set of experiments.

Referee #2

In their manuscript "Live imaging of wound angiogenesis reveals how macrophage phenotypic state orchestrates vessels sprouting and subsequent regression during repair", the authors primarily utilized the zebrafish model to show how macrophages at different phenotypic states contribute to wound neoangiogenesis and vascular regression in a wound healing scenario. The optical transparency of zebrafish embryos was fully utilized to live image how macrophages at different phenotypic states physically interact with growing blood vessels during wound repair. Of particular importance, the authors suggest that macrophages have differing roles in angiogenic repair depending on their inflammatory states. *tnf*-positive Inflammatory macrophages express *yrk* and *cyba1* proposed to repel anti-angiogenic *sflt1*-expressing neutrophils, and express *vegfaa* proposed to stimulate wound neoangiogenesis, while *tnf*-negative macrophages induce vascular apoptosis and phagocytose endothelial cells to promote vascular regression.

Overall, this research has revealed novel mechanisms for wound neoangiogenesis and vascular regression during wound healing that may have important clinical relevance. The data are mostly clear and convincing but there are a few points that need further evidence and areas that should be improved before publication.

Major comments:

1) In the zebrafish: the QPCR data suggesting that *Vegfaa* is responsible for regulating angiogenesis in wounds is supported by treatment with a single chemical inhibitor only. In the absence of a tissue specific loss of function study for macrophage *Vegfaa*, further cumulative evidence is needed to support that *Vegfaa* is responsible for wound neo-angiogenesis. It would be helpful to see the macrophage expression of *Vegfaa* *in situ* in the *TNF α* positive macrophages. It would also be necessary to show that multiple *Vegfr2* signalling inhibitors give the same effect to be confident of the specificity of that result. More convincing would be to perform the wounding assay in the *vegfaa* genetic mutants recently reported by Rossi et al (2016) which can be rescued for early

developmental defects by simple mRNA injection and would thus serve as a workable model to test this hypothesis in a mutant setting.

These are all good suggestions that we agree will strengthen our paper. We have now performed in situ for vegfaa using a new super sensitive protocol from RNAscope, and these show higher vegfaa expression in the wound site when macrophages are present, and specifically within tnfa positive macrophages in the wound area (see Fig 6B). We have used another vegfR2 inhibitor, su5416, and this also blocks wound angiogenesis (Sup. Fig. 6). Finally, we have imported vegfaa mutants from the Stainier lab, rescued via vegfaa mRNA injection as previously published (Rossi et al), and performed wounding experiments on these fish. We find that whilst recruitment of macrophages is normal, there is no subsequent wound angiogenesis (see Fig 6D), supporting our thesis that macrophage derived vegfaa is the sprouting signal.

2) Although it is clear that macrophages are required for vascular regression, the link between macrophages and vascular apoptosis is still weak. Please show whether tnf-negative macrophages at 10DPI express more macrophage paracrine factors that were shown to induce vascular apoptosis such as wnt7b, ang2 or similar when compared to tnf-positive inflammatory macrophages. It would also strengthen claims to show whether macrophages isolated from unwounded fish express less of these paracrine factors (as macrophage driven vascular apoptosis may be less frequent in unwounded fish).

We agree that further examination of the link between macrophages and vascular apoptosis would strengthen our paper. Wnt7b and ang2 are sensible suggestions but we see no significant expression of either of these in wound macrophages, whether tnfa negative or positive macrophages (which we now report on page 11), so it appears that a different mechanism is involved, but we feel that identifying what this signal might be is beyond the scope of this paper.

3) For all of the data presented, the use of bar graphs and the absence of individual datapoints in graphs weakens the accessibility of results. The paper would be improved by showing dot-plots throughout to make the individual n-values for each experiment more obvious.

We have now re-processed our data accordingly for the appropriate bar graphs.

4) Figure 1B: The authors have shown that wound neoangiogenesis in mice becomes more organised and returns to a distribution closely resembling unwounded skin by 14DPI. Although the quantification of blood vessel coverage indicates this, images in Figure 1B show that the blood endothelial cells in 14DPI mice are still disorganised. The manuscript would be improved by quantifying organisation of the vessels.

Reviewer 1 has also pointed this out. We have provided this quantification and supplied a more representative 14dpi image.

5) Figure 1D, 7A: How often do you see macrophages with engulfed endothelial cell material? This should be quantified in both mice and fish.

In our mouse studies we see 19.3% (SD 3.8%) of macrophages with some red CD31 stained endothelial material within them at the peak of resolution, 10 DPI. In our zebrafish studies the percentage of macrophages with engulfed material appears much less (1.2%, SD 0.7%) at what we think is the peak time of resolution, 8 DPI. The reason for this difference is unclear except of course far fewer cells are involved in the damage response in the fish model, therefore far fewer endothelial cells to clear. These data are reported on pages 4 and 11.

6) The fish were raised to 10 days post-injury (14 days post-fertilisation). Please describe whether fish were fed after 6-7 days post-fertilisation (when yolk sac is largely depleted). If yes, please include this in the method section. If not, some of the data may require reproduction as starvation state of the fish may significantly skew the results observed.

The fish were indeed fed from 6 days post fertilisation, and fish were rejected that had not grown to at least 5mm in length by 10 DPI from the analysis. We have included these details in the methods section, as suggested.

7) It is quite clear from the Movie 1 that neutrophils are only drawn to the vascular tip for a short period of time. Similar to the one done with macrophages, it would be helpful to quantify the percentage of sprout tips associated with neutrophils.

Most vessel tips are visited by neutrophils in our movies, but only transiently. They are rarely caught in direct contact with vascular sprouts in our stills following large scale needle stab wounds, which is where we quantified percentage of sprout tips associated with macrophages (Figure 2G).

8) The tnf-positive Inflammatory macrophages had been shown to only interact with vascular tip cells during neoangiogenesis and at later stage (4DPI), this is replaced with tnf-negative macrophages. Please indicate whether the tnf-negative macrophages are tnf-positive Inflammatory macrophages that switched phenotype or whether fresh tnf-negative macrophages are recruited to the site of injury over time.

This is a very interesting question, that we have also been asked several times when talking about this work at meetings. We can say definitively that we have seen cells expressing GFP (ie tnfa positive) diminish in their GFP positivity during the period of our imaging and this was not due to bleaching because other nearby cells remained GFP bright (see Sup. Fig. 4F and Movie 9). We now report this on page 8 and discuss this briefly on page 14; however, we cannot exclude also that some of the tnfa negative cells at the wound site were newcomers that had never previously been tnfa positive at the wound site. The whole issue of whether the tnfa positive to negative wave is due to changes within individual cells or influx of new populations is fascinating and we can briefly discuss here but we would like to follow up more carefully in a fuller study using better tools for tracking cells and their expression profiles; zebrafish, because of their imaging opportunities, will clearly be a good model in which to do this.

Minor comments:

1) It would be nice to see the correlation between macrophage number and wound recovery plotted as a correlation and supported by an r2 value.

We're not exactly sure what is meant here. Wound vessel pattern is generally re-established before the last macrophages have left the scene; if macrophage numbers are depleted to below a threshold of 20% (in fish) at the early stages of the repair process (as indicated by Sup Fig. 3C-E) then the vessel pattern does not re-establish...

2) The QPCR mFlt1 negative data should be shown. Was there a positive control for that experiment?

The positive control for this experiment was qRT-PCR performed on whole uninjured fish at 5DPF, showing expression of both Flt1 variants in the larval fish during growth and development. We have included this data as requested (Sup. Fig 3F).

3) Can you comment whether the repaired vasculature roughly resemble unwounded vasculature? If so, there may be guidance factors that direct neoangiogenesis along pre-determined routes.

Yes, the fully repaired vasculature in fish very closely resembles unwounded in pattern. We speculate that this may have to do with signals and physical space afforded by the muscle tissue and somite architecture; such cues are not present in the healing mouse wound and this might explain why the mouse doesn't return to exactly the pre-wounded state (see response to earlier Reviewer 1 comment).

4) Figure 2G: does the author mean percentage of sprout tips with associated macrophage?

Yes, that is what is meant in this graph – we have clarified this in the figure label.

5) Page 6: macrophages, "neutrophils are retained at the wound site and a maintain a long" should be "and maintain a long".

Apologies, we have made these corrections.

6) Similar to Tazuin et al., 2014 (reference 33), did you see contact-dependent neutrophil reverse migration in your wound neoangiogenesis model? Were these neutrophils associated with the vascular tip before making contact with macrophage?

Upon further examination of our movies, it does appear to be the case that most neutrophils leave the vessel tip after a contact with a recruited macrophage, but of course this is just correlative; we now mention this in the text (page 6).

7) Figure 5C: If proportions of macrophage were quantified, would the total number of proportions roughly add up to 10? Please clarify how these proportions were calculated.

Macrophages were scored for whether they were associated with either vessel sprouts, their sides, or at sites of anastomosis, or not associated with vessels at all (the majority of macrophages). Each category was then determined as a percentage of total macrophages counted per co-culture experiment. We have changed the graph axis to read "percentage of macrophages".

8) Figure 7C: figure labels need fixing

Apologies, we have corrected these labels.

Referee #3

Gurevich et al. show that macrophages are essential for wound angiogenesis and associate with new sprouting blood vessels. They find that pro-inflammatory macrophages in particular play a role in initiating and driving vessel sprouting, at least in part via the release of VEGF-A. Additionally, macrophages also affected wound angiogenesis indirectly by dispersing anti-angiogenic neutrophils that are recruited in the early stages of wound repair. Finally, the authors demonstrate that macrophages undergo a switch to an anti-inflammatory phenotype during the resolution phase and mediate blood vessel regression.

Others have already reported the involvement of macrophages in developmental and physiological angiogenesis, as well as in wound angiogenesis. According to the authors, the dynamic nature, precise function of the macrophage phenotypes, exact timing of interaction and roles of other leukocytes such as neutrophils remained to be investigated. To address these questions, the authors mostly used real time in vivo imaging in zebrafish.

The time course of macrophage and neutrophil recruitment has already been studied in adult zebrafish tail fin regeneration (Petrie et al., *Development*, 2014). Also, the kinetics of macrophages recruitment after wounding and infection with *E. coli* in zebrafish has been previously reported by Nguyen-Chi et al. They showed that unpolarized macrophages are recruited to the site of inflammation and start expressing TNF- α . Afterwards they are converted into the M2 phenotype during the resolution phase (Nguyen-Chi et al., *eLife*, 2015). However, both publications did not study the effect on blood vessels. This is where the present manuscript provides novel and relevant insights.

Although most of the findings in this study are well supported by the data, existing issues with the used transgenic zebrafish lines lead to the fact that some of the conclusions are not very convincing. Previous experience by several labs with the Tg(*fli1a:eGFP*), the Tg(*fms:gal4-VP16*);Tg(UAS:*nfsB-mCherry*);Tg(*fli1a:eGFP*) and the Tg(*mpeg1:gal4*);Tg(UAS:*nfsB-mCherry*);Tg(*fli1a:eGFP*) lines, demonstrate that macrophages are sometimes labeled by the *fli* promoter and at the same time not labeled by the *fms* or *mpeg* promoter. (This can even be seen in some of the presented movies, see Movie 6.) (A) For the Tg(*fli:eGFP*);Tg(*mpx:eGFP*) line with or without additional transgenes for macrophage labeling, this means that it is possible that the green labeled cells are not neutrophils,

but macrophages. On the other hand, the macrophages are sometimes also double labeled (green by the *fli* promoter in addition to the red signal by *fms* or *mpeg* promoter). **(B)** For the *Tg(fli:GFP);Tg(mpeg:mCherry);Tg(tnfa:GFP)* line, this means that one would be led to assume that these macrophages are TNF- α positive, while this might not always be the case. **(C)** For the *Tg(fli:GFP);Tg(mpeg:mCherry)* line, the authors cannot conclude with full confidence that the double labeled macrophages contain phagocytosed vessel material (although this may be likely, even if not mechanistically important). **(D)** Additionally, it is important to note that there are often gene silencing issues with UAS transgenes. As a result the fraction of the macrophages labeled or not will vary considerably, in particular when breeder selection is not carefully managed. Secondly, the claim that neutrophils only appear early and transiently at the wound site, although interesting and potentially relevant, is not really supported by the data (see specific comments for Fig. 3c, Movie 1, and Movie 3).

(A) Neutrophils and macrophages are very different in morphology and behaviour. Furthermore, the cells we identify/report as neutrophils in these movies express extremely bright GFP, and are over-exposed so as to enable us to see the much weaker signal of *fli:GFP*, which in turn is many times brighter than the macrophage red signal – we suggest that it would be highly unlikely that a macrophage could transition from being weakly red to very strongly green. But as further confirmation we have now utilised a *Tg(kdrl:mCherry-CAAX)* line which marks blood vessels in red (*kdrl* is the other *vegf* receptor), and in this complementary study, the only possible green cells are neutrophils. The movie we show is of a shorter timecourse but higher temporal resolution and confirms again the early neutrophil (green) presence and subsequent macrophage domination at the vessel repair site. (Movie 5 and stills shown in Sup. Fig 2E).

(B) As for the previous point, we have supplemented our existing movies and images again with a *kdrl* cross to our *mpeg* and *tnfa* fish, which means that yellow fluorescence can only be due to *tnfa* upregulation in macrophages. These new movies show similar macrophage *tnfa* expression kinetics to our earlier experiments using *fli* (Movie 9, Sup. Fig. 4F).

(C) As with Referee 1s concerns regarding our mouse studies, we are suggesting that macrophages are not dual labelled, but rather, the macrophage contains a phagosome with contents including green labelled vessel material. We have now included an image which demonstrates this more clearly (Figure 7A).

(D) It is true that some *Gal4/UAS-nitroreductase* based experiments might not represent completely clean deletions because of gene silencing issues, which is why we have complemented these studies with clodronate killing experiments and we were pleased to see that both approaches gave similar results (Figure 3A, B, Sup. Fig. 3A, B).

Furthermore, the manuscript leaves a couple of questions unanswered. **(A)** Do macrophages facilitate wound angiogenesis via other mechanisms besides VEGF signaling? **(B)** The used methods to interfere with VEGF effects are not selective for cell-type derived VEGF and there is no direct evidence for macrophage derived VEGF being the main mechanism. **(C)** Are macrophages really driving EC apoptosis during blood vessel regression or are they just clearing apoptotic ECs? **(D)** What triggers their phenotype switch?

(A) We provide several lines of evidence using both zebrafish (Figure 6) and tissue culture (Figure 5) approaches that blockage of *vegf* signalling is sufficient to induce a failure of wound angiogenesis/sprouting angiogenesis. This presumably means that other signalling pathways are largely dispensable for directing angiogenesis although, of course, there could be some redundancy.

(B) It would be difficult to demonstrate that the *vegf* driving wound angiogenesis is derived only from macrophages because lineage specific gene knockdown is still very hard in zebrafish, but our tissue culture experiments address this, in part, since here we are able to show that blockage of *vegf* from inflammatory macrophage conditioned media is sufficient to block most sprouting angiogenesis. Also our new *vegf* in situ (Figure 6C, see comment 1 from Reviewer 2) indicate that one population of wound macrophages express *vegf*.

(C) We still have no definitive answer here. To this end, we have crossed the macrophage transgenic line onto the kdrl/seca5 double transgenic that we used to identify endothelial cell apoptosis in wounds, and we see reduced numbers of apoptotic cells after clodronate injection to delete macrophages (Figure 7D, E), but we do not see expression of wnt 7 and other signals shown to mediate macrophage killing of cells in other models, so we are not clear about mechanism in this situation – we now mention this in the text, page 11.

(D) What triggers the phenotype switching we observe in wound macrophages is clearly an important question. All we know is that the switch and its duration can be modulated as we have demonstrated by addition of exogenous factors, but what are the natural endogenous signals is not tractable in the current study; indeed, it might be the case that the switch is a passive occurrence that happens 24-48hrs after initial activation once reaching the wound. We have added a little speculative discussion on this topic in the Discussion (page 14).

Curiously, the authors suggest that the pro-inflammatory macrophages are the major source of VEGF, whereas anti-inflammatory macrophages produce less. To this reviewer's knowledge, mouse studies rather suggest that the anti-inflammatory macrophages of the M2 polarization (characterized by MRC1 expression) are a major source of VEGF.

The literature is mixed on this; as Reviewer 1 pointed out, there are certainly plenty of murine papers showing that pro-inflammatory macrophages can express vegf, just as we do in this fish study. However, this reviewer is right, there are others that show the contrary. Our initial manuscript mentioned this potential discrepancy in our introduction (page 2), but our own data is very clear cut in that vegf expression is primarily by the earliest incoming, pro-inflammatory macrophages.

Overall, this work is very interesting, but requires some improvements concerning the issues with the transgenic lines. Providing more convincing movies (e.g. from other transgenic lines), will largely address the major concerns. However, there are other issues with the work that requires some attention, including statistics. I listed below a number of specific comments and additional non-essential suggestions for the authors will hopefully find useful for improving their interesting study.

SPECIFIC COMMENTS

Statistics

The statistical analysis presented is problematic. Normality of the data should be checked. If the data are not normally distributed, the authors need to transform them or use a non-parametric test. If the data are normally distributed and only have two independent groups, the t-test can be used. If there are more than 2 groups (which is the case for Fig. 3b; Fig. 4f; Fig. 5c,e,f; Fig. 7e,g,h,i; Sup. Fig. 2e, Sup. Fig. 3d and Sup. Fig. 4a), t-test is not acceptable. The authors should use ANOVA and correct for multiple testing to reduce the number of false positives.

As suggested we have checked for normality of distribution for all data and retested the instances of 2 or more groups by one-way ANOVA, which was post-tested via Bonferonni comparison, a process that corrects for multiple testing.

Figures

Figure scale bars: sometimes the scale bar is present on every picture, sometimes only on 1 or 2 pictures or even missing completely. More consistency here would be useful.

Apologies, we have corrected for this.

Figure 1: (b) The picture of 14 DPI looks very different from the picture of the unwounded skin, although the authors describe in the text that it resembles unwounded skin.

Reviewers 1 and 2 also noticed this (see our earlier response), and we have presented an image that more closely resembles unwounded skin; our quantification in Figure 1D suggests that at this stage the pattern of vessels will not be completely back to unwounded, although vessel density largely is (Figure 1C).

(c) Maybe quantifying other characteristics like vessel length and/or complexity like the authors did for fish later on would be useful, because now the quantification of blood vessel coverage for unwounded skin and wounded skin at 14 DPI is almost the same, but the pictures are not similar. The picture of 0 DPI does not seem representative for 20% blood vessel coverage.

We have now performed an additional analysis of vessel alignment which we hope helps here. Since we always include a little margin of “unwounded” tissue our quantification will always be a slight over estimation of wound vessel metric.

(d) Using a lower magnification would give a good overview of the different types of interaction. The authors could also stain for tip cells (e.g. ESM1) to really visualize that the macrophages are associated with tip cells in the early stage of wound angiogenesis.

We now have included a lower magnification view of a late stage wound (Sup. Fig. 1A) that we hope indicates the complexity of interactions in these tissues and complements the high mag views we show also in Figure 1E.

(e) The text is not really describing what can be seen from the graph. According to the text, the macrophages are associated with tip cells from 3-7 DPI and show a "hugging" behavior from 7-10 DPI. However, the percentage of macrophages associated with sprout tips at 7-10 DPI is not very different from 3-7 DPI.

The Reviewer is right, there are similar numbers of associations (we have changed the y axis to use term “association” for clarity now) at these two time windows but the nature of the interactions skews from “tip association” to “hugging” behaviour. We have tried to be more explicit about this in the text, page 3.

Figure 2: (a) It should be "wounds" instead of "wouns" in the figure legend.

Oops, thanks.

(b) What do the different colors mean in the graphical representation of the nodes and sprouts?

The colours are simply the way that the AngioAnalyser plugin classifies the various elements that it measures ie different colours for vessels deriving from more complex, higher order, branchpoints/nodes. We now flag this up in the Legend. We would rather keep this but we don't specifically use this additional information in our vessel analysis and so could change to grey scale.

(d) The authors can maybe indicate in the first picture that the fish is not wounded yet, and include a box that denotes the wound site. Maybe also include a picture of 12 HPI, since this is an important time point, namely the peak of neutrophil recruitment.

Yes, we have made these changes including addition of a 12HPI image.

(f) To make it more complete, the authors could also include pictures of neutrophil recruitment.

We left neutrophil images out of Figure 2 in order to focus here on macrophages, but have reinserted now.

(g) Label of the Y-axis of the right graph is incomplete.

Thanks – now corrected.

(i) Scale bar is missing at the left picture. It is not really clear that the vessels have anastomosed and are perfused at 930 MPI. It seems that this is happening outside the range of the z stack. Therefore, one cannot conclude from these pictures (and movie) that the sprouting only happens in a ventral to dorsal direction. In addition, the authors might want to use a transgenic line in which the neutrophils have a different color, for example labeled with CFP, because with this transgenic line, one cannot be certain that the green labeled cells are neutrophils. If the authors would use a third color, they can

easily distinguish between ECs, macrophages and neutrophils. Moreover, it would be easier to differentiate ECs from neutrophils that are localized around blood vessels. (k) Similar comment as for (i), from these pictures and movies, one cannot conclude when anastomosis and perfusion are precisely happening.

Scale bar now added. We are confident that our z-stack captures anastomosis and perfusion events in their entirety as well as their directionality – to this end, we have added a 3D projection corresponding to the 930 MPI timepoint showing the entire z-stack (Movie 4). Neutrophils can be distinguished by their extremely bright GFP expression relative to any fli positive cells, and their very different behaviours. However, we have now supplemented these movies using kdrl:mCherry-CAAX fish to mark blood vessels red, which makes for easier colour based differentiation of ECs from neutrophils (green) and specifically demonstrates that neutrophils do not dwell at damaged vessel tips in these wounds (new Sup Fig 2E). With respect to Figure 2K, these are pictures of dextran loaded vessels only (not movies); we use them to highlight that by 1440MPI, anastomosis and perfusion have happened, rather than making a claim as to precisely when this occurs (we believe that Movie 2 shows this).

Figure 3: (b) Please make the bar labels less confusing, maybe by mentioning (part of) the transgenic line that was used. Please be consistent in the colors of the bars: keep the same color for "Met treated ctrl injury" in Sup. Fig. 2e and Fig. 3b.

We have adjusted the graphs as suggested.

(c) Same comment as for Fig. 2i about the transgenic line. One can see neutrophils here in the control embryo at 230 MPI, which is contradictory with the statement that neutrophils are only present from 30 to 90 MPI.

We hope that our inclusion of the kdrl line marking vessels in red, as discussed above will allay this concern. Our statement that neutrophils are only present from 30 to 90 MPI (on average) is relevant to the smaller, partial ablations presented in Figure 2, not to the full vessel ablations presented here. As the graph in E shows, neutrophils in this context persist for longer, given the larger injury and increased inflammatory response.

(d) It would be useful to try to keep the labeling consistent, rather use "Untreated injury" and "Macrophage Met ablated injury" like in b instead of "Untreated control" and "Metronidazole treated injury", and try to use the same bar colors as in b. (f&g) Rather use "Wounded" instead of "Wound" in the bar labeling.

We have adjusted the labelling as suggested.

Figure 4: (b) The first picture is not representative for 50% of TNF- α positive macrophages like described in the text and visible in e. (In e the number is even higher than 50% for DMSO ctrl.) The third picture is also not representative for the percentage in e. (d&e) Include "WT" in the graph legend, like in Sup. Fig. 3b&c.

This is likely an issue with the relative colour level of GFP versus mCherry, with cells that are only slightly GFP positive appearing as mCherry only. We have now added a green only, single channel image for each treatment, to support our quantification (Figure 4B).

(g) Same comment as for Fig. 2i: it would be more ideal to use a different color for TNF- α signal. In Tg(fms:gal4-VP16);Tg(UAS:nfsB-mCherry);Tg(fli1a:eGFP) and Tg(mpeg1:gal4);Tg(UAS:nfsB-mCherry);Tg(fli1a:eGFP) lines, the macrophages are sometimes double labeled (green by the fli promotor in addition to the red signal by fms or mpeg promotor). In this case, one would assume that the macrophages are TNF- α positive, while this might not always be the case. In the picture of 30 MPI, the arrow on the right seems to indicate a TNF- α negative macrophage instead of a TNF- α negative one. From these pictures, it is not really clear that TNF- α positive macrophages are interacting earlier and more frequently with sprouting vessels than TNF- α negative ones.

Unfortunately, the TNFa is a BAC, so cannot be quickly re-engineered, as a minimal promoter line could be. But now we have complemented these studies with a line expressing the

kdr1:mCherry CAAX (Sup Fig 4F, Movie 9), so that any macrophages that are co-labeled with GFP can only be tnfa positive pro-inflammatory ones. The macrophage in question is expressing the TNFa reporter initially but at low level and this increases with time. Most of the early interactors are indeed TNFa +ve.

Figure 5: (a&b) What are the small red dots? Did the staining not work properly? DAPI labels all nuclei, not only fibroblasts? Why are there so few macrophages present?

The red dots are background staining – macrophage staining using our chosen antibody (MCSFR) is notorious for not being particularly clean. This antibody was selected specifically to not cross-react with endothelial cells and fibroblasts, as many other macrophage antibodies can do. DAPI does indeed mark all nuclei, but it is the only marker we have in this assay that marks fibroblasts, hence the figure label. These are high magnification images showing independent interaction events, hence the number of macrophages.

(e) Description of the second and third graph is missing in the figure legend.

These graphs have now been moved to Sup. Fig 5B. More detailed descriptions of these have been added to the relevant figure legend.

Figure 6: (a) Description of second and third graph is missing in the figure legend.

Again, we have added more detailed descriptions to the figure legend.

(c) Mistake in the figure legend, it should be: "..., measured from images represented in B." (d) Should it not be 4 DPF instead of 4 DPI above the pictures?

The reviewer is correct, we have fixed these errors.

Figure 7: (a) Same comment about macrophage labeling by the fli promotor. It might be that the macrophages are green, just because they are labeled by the fli promotor, and not because they contain phagocytosed EC material.

We have replaced this image with one that demonstrates macrophage engulfment of green labelled endothelial cell matter more clearly (Figure 7A).

(b) Please be consistent in the name of the transgenic line. In the text you use "Tg(flk1:DsRed)" and in the figure legend you use "Tg(kdr1:DsRed)". Please indicate more clearly whether the picture came from a wounded or unwounded embryo.

Yes, now we have made more clear.

(c) Part of the labeling at the x axis is missing. (d) Again, please be more consistent in the names of the transgenic line. (g) Part of the y axis label is missing. Indication of statistical significance is missing.

Thanks, we have now amended.

Supplemental figure 1: Scale bars are missing. (a) inconsistent panel labeling: the authors could use "0 HPI" and "1 HPI" instead of "immediately post stab injury" and "one hour post stab injury".

Yes, we have now done as suggested.

(b&d) Please include a box that denotes wound site.

B shows images that are within the wound site, so such a box isn't practical. We have added a box to D to show the initial wound site.

(d) Same comment as for figure 2i: the authors might want to use another color for the neutrophils, especially in this double transgenic line, since one cannot know whether green labeled cells are

neutrophils or macrophages labeled by the *fli* promotor.

The point of this movie is to show the dynamic nature of vessel sprouting in a wounded context; the *mpx:GFP* is therefore a distraction in this regard. We have therefore shortened and cropped the movie to include only the vessel dynamics.

Supplemental figure 2: (a) Please be consistent in the names of the transgenic line, use either Gal4FF or KalTA4.

Yes, we have done as you suggest.

(e) At what time point is the total vessel length quantified? This is not clearly indicated in the figure or figure legend. The authors could maybe indicate more clearly that "Met treated control injury" are Tg(*fli:GFP*) embryos treated with metronidazole.

Vessel length was quantified at 4DPI. We have added this detail to the figure legends as requested.

Supplemental figure 3: (a) Please indicate what the colors represent (macrophages, TNF- α positive macrophages, etc.), like done for other pictures. (d) Include "WT" in the bar labeling like in Fig. 4f. (e) Same comment as for a. Why not show the green signal here, like in the other Dextran pictures (e.g. Fig. 2j)?

We have re-labeled images as suggested. Unfortunately, we cannot add the green signal here, as we did not image the green channel for this experiment – the focus of this panel was to demonstrate increased leakiness when vessels are treated with pro-inflammatory stimulants. As the reviewer points out, we show imaging of *fli* and dextran together previously (Figure 2J).

Supplemental figure 4: /
Movies & tables

The names of the movies files are confusing. In addition, all movies were uploaded twice. Please also include scale bars in the movies.

Apologies, the movies were uploaded in two different formats, to allow for compatibility with both Mac and PC systems. We will stick to one format (.AVI) this time.

Movie 1: Similar comment as for figure 2i: in this movie, the anastomosis and lumenization at 600 MPI as described in the text is not visible. It seems that this is happening outside the range of your z stack. Therefore, one cannot conclude from this movie that the sprouting only happens in a ventral to dorsal direction. What is the big red shape at the upper right? Unlike what is claimed in the text about neutrophils appearing early and transiently at the wound site, one can see several neutrophils interacting with the vessels around 720 MPI.

As above, we have included a rotating 3D projection corresponding to the final timepoint of this movie to demonstrate that we have indeed captured the entire anastomosis event (Movie 4). Sprouting happens in both ventral-dorsal and dorsal to ventral directions, but for partial ablations it generally occurs in a ventral to dorsal direction – we have stated this on Page 3. The big red shape is likely a dendritic cell. The neutrophils observed at 720 MPI do not engage with the repaired area (denoted by consistent macrophage contact).

Movie 2: It would be interesting to show this in the triple transgenic line that was used in Movie 1. Again, the same comment as for Figure 2i and Supplemental figure 1d about an extra color.

As above, the purpose of this movie is to demonstrate the dynamic nature of vessel sprouting. We have numerous other movies to deal with the immune cell-endothelial cell interactions. We have removed the initial part of the movie that has distracted from our main point.

Movie 3: Again, same comment as for movie 2: there are still neutrophils interacting with the sprouting vessel at 370 MPI.

The main phase of neutrophil-vessel interaction phase for this movie is at approx. 250 MPI – neutrophils seen at 370 MPI are no longer in the same z plane as the repairing vessels and are moving away.

Movie 4: /

Movie 5: In this movie, it is not really clear that TNF- α positive macrophages are interacting earlier and more frequently with sprouting vessels than TNF- α negative ones. Using a different color for the TNF- α positive macrophages might solve this problem.

As above, unfortunately changing the TNFa colour is not possible; however, we have provided further movies with the kdrl mCherry line to substantiate our claims here.

Movie 6: /

Table 1: Table 1 was included 2 times.

Apologies, we have removed one copy.

NON-ESSENTIAL SUGGESTIONS

• M&M: "Sham wounded fish were used as an uninjured control." Please clarify what is meant with "Sham wounded", because the fish are uninjured?

That's correct, fish are anaesthetised but uninjured.

• Second paragraph of the introduction: "Blood vessel regression during post-natal retinal development is also driven by macrophages, as their absence completely abolishes this process (8-10)." This statement is incorrect as the cited work refers to the specific case of regressing hyaloid vessels. There is no evidence for this process in development regression withing intrretinal vasculature.

Thanks, we have altered the text, pages 2 and 13.

• Typing/spelling errors:

o Timecourse vs time course

o One hour vs 1 hour

o Microlaser vs micro laser

o Quantitative PCR vs qRT-PCR

o Please use μm instead of um and μg instead of ug.

o In vitro and in vivo should be always in italics.

o Nomenclature of transgenic lines: a semicolon should be used between two transgenes instead of a comma and the name should in italics.

o Abbreviations: the first time, the full word used with the abbreviation between brackets, afterwards only the abbreviation should be used

Thank you for submitting your revised manuscript. The study has now been re-reviewed by referees #1 and 2. Referee #3 was not available to review the revised version.

Both referees appreciate the added data and explanations, and support publication here. The referees raise a few comments that I would like to ask you to resolve in a final revision - most of these are text edits. Referee #1 wants quantification of the data for the new figure 2B. Referee #1 also finds that the mouse data doesn't add so much in novelty but I find that dataset very informative and would like to leave it in.

REFEREE REPORTS

Referee #1:

Referee #1

The paper by Gurevich et al. investigates the influence of macrophages on the development of blood vessels following wounding. The authors use several animal models, such as mouse skin and embryonic zebrafish in addition to a HUVEC cell culture system to analyze the effects of manipulating macrophage numbers (in zebrafish) or their gene expression profiles on neo-angiogenesis. They define three different stages of macrophage involvement in mice: first macrophages are associated with tips of new blood vessel sprouts, macrophages subsequently "hug" new blood vessels and finally engulf material from dead endothelial cells. The next experiments are being conducted in zebrafish embryos, where the authors use either a needle punch or a laser to induce a wound. Here they first investigate blood vessel formation followed by an analysis of macrophage and neutrophil dynamics. They then study how an absence of macrophages influences neo-angiogenesis before manipulating the activation state of macrophages. In cell culture and zebrafish embryos, the authors finally provide evidence that macrophage-derived VEGF might be the pro-angiogenic stimulus regulating blood vessel growth after wounding. In general, the study combines 3 different systems for studying the influence of macrophages on blood vessel growth. The mouse data (see below) and the cell culture data clearly need more in depth analysis. The more detailed analysis has been carried out in zebrafish, but it is not clear at present how the mouse and cell culture data would add to this. For the zebrafish part, the data are of high quality with adequate quantifications provided.

Major:

1. The authors claim that in mouse wound blood vessels become "more organized and returning to a distribution closely resembling unwounded skin by 14 DPI" (page 3). This statement cannot be supported by the panels provided in Figure 1B. The blood vessels at the 14 DPI time point do clearly look very different (e.g. in caliber and organization) from the unwounded condition. Evidently, a more detailed analysis on this part is necessary.

We have now replaced the 14 DPI image with a more representative one (Fig 1B) and have quantified the angiogenic response and its resolution by analysis of vessel orientation (Fig 1C). At 14 days the vessels appear to resemble those seen in unwounded tissue in amount and organisation more closely than at early repair stages; however, they are not completely back to the unwounded pattern and we have softened our statement here (Page 3).

It is still not clear to this reviewer how the analysis of blood vessels in the skin punch wounds would add to the authors' claims. There have been numerous papers on blood vessel development in skin wounds (e.g. Urao, DiPietro, MicroCT angiography detects vascular formation and regression in skin wound healing, Microvascular Research, 2016) that have performed a more in-depth analysis of the vasculature during wound healing. Therefore, as it stands at this moment, figure 1 A-D does not provide novel insights.

2. The authors use CD31 and CD68 to differentiate macrophages from endothelial cells and claim

that in the clearance phase the staining of CD68 positive macrophages signifies the uptake of EC material. A recent publication shows that macrophages in skin lesions can express CD31 (Tidwell, Googe, *Journal of Cutaneous Pathology*, April 2014). The authors need to devise additional means to show an uptake of EC material by macrophages in order to substantiate their claim.

It is indeed the case that CD31 can stain macrophages as well as vessels in the manner indicated by the cited paper, but this doesn't seem to be the case for us. Only a subset of macrophages co-stain with CD31, and they do so in discrete domains within the cell (probably a phagocytic vacuole) - we have included a projection through the 'Clearance' macrophage in Figure 1E to demonstrate this (Movie 1). We have also included an expanded view (Sup. Fig 1A) showing how only a subset of macrophages (green) co-stain ex-vessel material (red). And to be sure, we have also supplemented these data with new frozen sections where we have stained vessels with VE-Cadherin antibody, a marker of endothelial cells that is not expressed by macrophages; again, we observe ex-vessel material in macrophages (Sup Fig 1B). Our new TEM data (see later) is also supportive that at these later stages, assorted macrophages have engulfed cell debris near vessels.

OK.

3. In order to show a conserved role of macrophages during neo-angiogenesis in mice, the authors need to deplete macrophages in mice, e.g. via clodronate treatment or use of CD11b-DTR mice, and analyse blood vessel development. Referring to my point 1, the mouse data lack sufficient depth of analysis at this point.

The murine clodronate ablation experiments suggested here by Reviewer 1 have already been performed in two previous publications and shown to alter vessel patterning following wounding (Lucas et al, 2010, Willenborg et al, 2012), as we have mentioned in our manuscript (page 2). The objective of our study was to try to extend our mechanistic insight of this process by directly observing the cell interactions underlying inflammation mediated wound angiogenesis, by taking advantage of the genetic and imaging opportunities in zebrafish. The mouse data collected for our paper is primarily to descriptively expand our understanding of the vessel repair timecourse and to distinguish each of the various macrophage-vessel interactions seen throughout wound repair, rather than attempting to address any mechanistic questions. However, to improve the depth of analysis of the mouse data, we have now undertaken some TEM imaging of the various stages of wound repair (Fig 1E). This allows us to more subtly describe each of the macrophage/vessel interaction events we observe in mouse which we then follow up mechanistically in fish.

The authors' response is in line with my comment on point 1 (limited novelty of mouse data), also the time course of blood vessel development/regression after wounding has been described before.

4. The authors are investigating different aspects of blood vessel formation after wounding in zebrafish. If I understand the paper correctly, they investigate for one the formation of new blood vessels in places, where there were no blood vessels before (needle damage) and on the other hand, they analyse the "repair" of blood vessels that were damaged in their normal place (microlaser). These two processes might rely on different mechanisms and need to be characterized in more detail. The authors state that "sprouting during wound angiogenesis appears to follow a similar process to that reported for embryonic development of the ISVs" (page5). A recent report (Wild et al., le Noble F., *Nature Communications*) showed that later sprouts mainly emanate from veins. Is this also true during wound healing? The authors state that for small ablations vessel outgrowth generally occurs in a ventral to dorsal direction. Is this true for both venous ISVs and arterial ISVs? This would be important for comparing blood vessel sprouting during embryogenesis with sprouting during wound healing.

We think we may not have explained our various lesions clearly enough. We have made two categories of mechanical lesions: large wounds (using a 30G needle) that damaged vessels, versus small wounds (using a fine tungsten needle) that damaged tissue without puncturing a vessel. Both of this class of mechanical lesion trigger vessel sprouting which shows how vessel damage per se is not required for wound angiogenesis. The second class of lesion was more precise, made with a microlaser, to an individual vessel, and even these were not all the same. We made some mini lesions/cuts, so we could watch the rapidity of vessel tip sprouting and fusion, and we made some larger laser lesions where an intersomitic vessel was completely vapourised so we could observe

filopodial sensing as the vessel tips searched for one another from a distance. We have now more clearly spelled out the different wounding regimes in the text (pages 4 and 5).

OK.

The referee raises a very interesting question. Do the ingrowing vessels derive from vein or artery? Because of this query, we have now injured fish with fluorescently identifiable arteries versus veins (a gift from Sarah De Val, Oxford), and we see that sprouts generally emanate from whichever vessel is damaged/closest to the area of injury, and that both arteries and veins are capable of readily contributing to wound angiogenesis, often leading to chimeric vein-artery derived vessels in the wound (see new Fig 2B)

The data on sprouting from artery vs. vein look very interesting. However, they need to be quantified. In addition, the presence of dll4 positive cells in veins is not a proof that these cells came from the neighboring arterial blood vessel. It could also be that these cells turn on dll4 (that is GFP) expression, while being vein derived. Time-lapse movies to show sprouting from arteries and/or veins would be helpful to substantiate the authors' claims. The single panel in figure 2B is not very helpful. How many days post injury was this picture taken? How about other time-points? How about the resolution process? In addition, while the authors now state in the results that "In contrast to embryonic development where sprouting appears to proceed primarily from venous endothelium (Wild et al, 2017), we observed that both arteries and veins readily contribute to wound angiogenesis, resulting in chimeric wound vessels (Fig. 2B).", they still state in the discussion that "wound neoangiogenesis appears to largely recapitulate earlier episodes of embryonic developmental angiogenesis."

5. The top 5 parameters in Figure 2K and Figure 3E (control situation) should represent similar processes. Why are there big differences, e.g. in neutrophil presence at vessel (yellow bar), blood vessel sprouting or re-establishment of flow? In addition, Figure 2K is redundant to Figure 3E. Figure 4I again shows a similar representation of the control situation.

The differences are present in these metrics due to the variation in extent of lesion as discussed in response to comment 4 above, and as we now more fully indicate in the text (pages 4 and 5). Larger injuries elicit a larger inflammatory response, and take longer to sprout, anastomose and re-establish flow, and we have shown appropriate controls for each experiment. We have now more fully explained this in the text on page 5 and indicated which injury was performed in each experiment.

OK.

6. The cell culture experiments (Figure 5) do not provide novel mechanistic insights. For instance, it is well established that IFN gamma stimulation of macrophages increases VEGF expression (and thereby angiogenesis), e.g. Ramanathan, Giladi, Leibovich, *Experimental Biology and Medicine*, 2003; Xiong, Elson, Legarda, Leibovich, *Am. J. Pathol.*, 1998.

We think that our experiments do show additional insights; for example, our more extensive analysis has revealed novel observations regarding macrophage-vessel interactions (the increased localisation of macrophages to vessel tips after macrophages have been IFN gamma stimulated), and it is worth nothing also that our experiments utilise human primary cells (macrophages and endothelial cells), rather than murine cells as in the papers this referee cites. Reviewer 3 (see later) asks us for evidence that the vegf that drives wound angiogenesis is largely derived from macrophages and of course this is very difficult to address in vivo in zebrafish because of difficulties in lineage specific knockdown, but our reductionist, in vitro experiments, where blocking antibodies against vegf deplete the effects of macrophage conditioned media, go some way to addressing this. But your concerns led us to think of other novel insights that this co-culture approach might offer. We have now purified human neutrophils that we leave untreated or treat with a type 1 Interferon, IFN alpha (to skew neutrophils towards an inflammatory phenotype, Pylaeva, Lang and Jablonska, *Front. Immunol.*, 2016). When we apply these neutrophils to our in vitro angiogenesis assay, we find that neutrophils, and particularly inflammatory neutrophils, appear to suppress blood vessel growth (Fig 5I, J), just as indicated by our fish imaging approaches. This again demonstrates the power of such

complementary studies, and to the best of our knowledge is a novel set of experiments.

It would be good that the authors point out the fact that the macrophage-VEGF-angiogenesis axis has been established before. The neutrophil data are nice.

Note on methods: It would be advisable to add the line numbers for all transgenic lines used to allow the reader to more easily identify them (e.g. y1 for Tg(fli1:EGFP)).

Referee #2:

On re-reading the manuscript carefully, my view is that this study represents a very useful, mechanistic addition to our understanding of immune cell-angiogenic interplay. The zebrafish data are very strong and at the heart of the study. The new mutant data added and the improved quantification and controls throughout have significantly improved the study. Overall, the work is well performed and carefully controlled. The zebrafish findings are well supplemented with in vitro and some mouse studies but also stand alone as highly informative and novel.

The authors have provided a very strong revision. They have in most parts responded to the reviewer's queries with new data and have clarified all of the major points that I previously raised. I have no further major concerns and only list a few minor suggested edits below.

Minor suggested edits:

Page 4 - "In contrast to embryonic development where sprouting appears to proceed primarily from venous endothelium (Wild et al, 2017), ..." what specific embryonic development context are you referring to in the Wild paper?

Throughout: Check proper transgene names and nomenclature carefully - eg. Tg(fli:GFP) - is this Tg(fli1a:EGFP)?

Throughout: Wording is at times a little emotive, which can detract from clarity - eg. "...direct intimate associations..." what does this mean in real terms? And "...in a highly dynamic fashion apparently sensing directionality before..." Do you mean sensing direction, location?

Page 6 - "To complement this genetic approach..." is this better described as a chemical-genetic model? I would consider a genetic mutant devoid of macrophages to be a purely genetic model so the term seems inaccurate to me.

Page 8 - "... tnfa positive macrophages was also observed after vessel injuries performed on (kdr1:mCherry-CAAX); Tg(tnfa:GFP); Tg(mpeg:mCherry) triple transgenic fish..." It might be useful to add a sentence explaining why this specific transgene combination was used in the experiment being referred to. Currently, it reads like an odd add-on.

Page 11 - "Blocking all caspase activity in this manner mimics late stage macrophage ablation (Fig. 7B, C)." - Do you mean produces phenotypes that mimic those seen upon late stage macrophage ablation? This is a little unclear.

Page 11 - "FACS sorted" FAC sorted?

Please find attached our revised manuscript (also uploaded to the EMBO portal), we have highlighted all changes other than small nomenclature changes in teal.

We agree with Referee #1 with regards to the quantification of the new Figure 2B data, and have added this to the text (Page 4). We have inserted the references suggested by Referee #1, clarified the points throughout the manuscript raised by Referee #2, and fixed some minor nomenclature and labelling issues throughout the Figures raised by both Referees.

We have also re-ordered the document as per your instructions, and taken on board the editorial edits.

Corresponding Author Name: Paul Martin

Journal Submitted to: EMBO J

Manuscript Number: EMBOJ-2017-97786